# FROM MARKOV TO LAPLACE: HOW MAMBA IN-CONTEXT LEARNS MARKOV CHAINS

**Marco Bondaschi** [*]
EPFL

**Nived Rajaraman**
UC Berkeley

**Xiuying Wei**
EPFL

**Razvan Pascanu**
Google DeepMind

**Caglar Gulcehre**
EPFL

**Michael Gastpar**
EPFL

**Ashok Vardhan Makkuva**
Télécom Paris

## ABSTRACT

While transformer-based language models have driven the AI revolution thus far, their computational complexity has spurred growing interest in viable alternatives, such as structured state space sequence models (SSMs) and Selective SSMs. Among these, Mamba (S6) and its variant Mamba-2 have shown remarkable inference speed-ups over transformers while achieving comparable or superior performance on complex language modeling tasks. However, despite these architectural innovations and empirical successes, the fundamental learning capabilities of Mamba remain poorly understood. In this paper, we address this gap by studying in-context learning (ICL) on Markov chains and uncovering an interesting phenomenon: even a single-layer Mamba efficiently learns the in-context Laplacian smoothing estimator, which is both Bayes and minimax optimal. To explain this, we theoretically characterize the representation capacity of Mamba and reveal the fundamental role of convolution in enabling it to represent the optimal Laplacian smoothing. These theoretical insights align strongly with empirical results and, to the best of our knowledge, represent the first formal connection between Mamba and optimal statistical estimators. Finally, we outline promising research directions inspired by these findings. Code is available at https://github.com/Bond1995/Markov-Mamba.

## 1 INTRODUCTION

Transformers have been at the forefront of recent breakthroughs in language modeling, driving the AI revolution (Vaswani et al., 2017; Radford & Narasimhan, 2018; Devlin et al., 2018). Despite their empirical success, transformers suffer from high computational complexity, such as quadratic scaling in sequence length during training and linear cache size at inference (Gu & Dao, 2023a). To address these limitations, there is a growing interest in designing alternative efficient architectures among which structured state space models (SSMs) are the most prominent. In particular, Selective SSMs such as Mamba and Mamba-2, have achieved state-of-the-art results in various language modeling tasks, while greatly improving the inference throughput (Cirone et al., 2025).

Motivated by this success, there is tremendous interest in understanding the sequential modeling abilities of SSMs, especially that of Mamba. In particular, mirroring a theme that has been successful in unraveling fundamental mechanisms (e.g. induction heads) behind transformers (Makkuva et al., 2025; 2024; Rajaraman et al., 2024; Nichani et al., 2024; Edelman et al., 2024), a growing body of research explores Mamba through its in-context learning (ICL) capabilities (Grazzi et al., 2024; Halloran et al., 2024; Akyürek et al., 2024; Park et al., 2024). While these works reveal interesting insights about Mamba's ICL abilities vis-a-vis transformers, they are largely empirical in nature, and we currently lack a fundamental theoretical understanding of Mamba and its underlying learning mechanisms. We are thus motivated to ask:

**Can we systematically characterize the ICL capabilities of Mamba?**

---

[*]Correspondence to marco.bondaschi@epfl.ch.

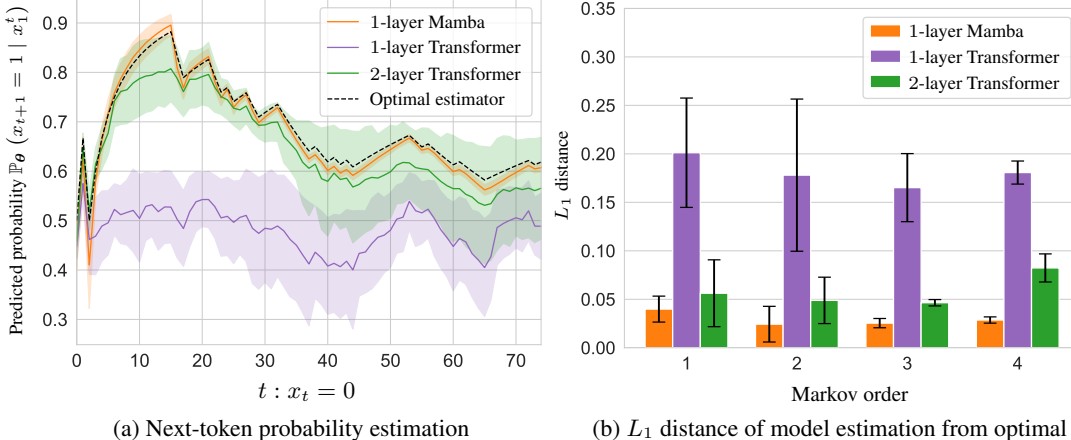

(a) Next-token probability estimation

(b) $L_1$ distance of model estimation from optimal

Figure 1: Single-layer Mamba learns the optimal Laplacian estimator when trained on random Markov chains, exhibiting ICL. (a) shows the predicted probability distribution on a fixed test sequence for models trained on binary first-order Markov sources. (b) quantifies the $L_1$ deviation from the optimal estimator for random sequences and various Markov orders. The error intervals show the standard deviation across 5 runs. Sec. 4.3 and Fig. 10 further discuss Mamba vs. Transformers.

In this paper, we approach this question from the point of view of representation power, and characterize Mamba's ICL capabilities on Markov processes, building upon the Markov-ICL framework originally introduced for transformers (Edelman et al., 2024). As opposed to a Mamba vs. Transformers comparison, here we leverage this framework for a detailed study of Mamba, and uncover an interesting phenomenon: even a single-layer Mamba efficiently learns the in-context Laplacian smoothing estimator, which is both Bayes and minimax optimal, for all Markov orders (Figs. 1 and 10a). Towards explaining this, we theoretically characterize the representation capacity of Mamba and demonstrate that the convolution mechanism, together with selectivity and recurrence, plays a fundamental role in realizing the Laplacian smoothing. Importantly, we showcase that these theoretical insights align strongly with empirical results, even outside the realm of Markovian data. To the best of our knowledge, this is the first result of its kind connecting Mamba and optimal statistical estimators.

In summary, we make the following contributions:

- Leveraging the Markov-ICL framework, we uncover the surprising fact that even a single-layer Mamba learns the optimal in-context estimator for all Markov orders (Fig. 1). Intriguingly, convolution plays a pivotal role, more so than gating and non-linear activation, in this learning ability (Sec. 3).
- Towards explaining this phenomenon, we characterize the representational capacity of single-layer Mamba and show, both theoretically and empirically, how it represents the optimal in-context estimator for any finite-state first-order processes, through an intricate interplay of convolution, selectivity and recurrence. Further, we provide fundamental limits for higher-order processes (Sec. 4).
- We demonstrate the generality of our findings on non-Markovian data and illustrate the fundamental role of convolution even on complex language-modeling tasks (Sec. 5).

## 1.1 RELATED WORK

SSMs (Gu et al., 2020; 2021) have been recently introduced as an alternative recurrent architecture aimed at rivaling the well established transformer backbone (Vaswani et al., 2017). The model was originally introduced as a discretized linear dynamical system (Gu et al., 2021). Recent works tried to re-frame the architecture from a linear recurrent perspective (Orvieto et al., 2023b). However, there are still many gaps in understanding this family of models (Team et al., 2024), such as questions around expressivity (Orvieto et al., 2023a). This is particularly important given the proliferation of Mamba-inspired architectures that have emerged since its introduction (Qin & Liu, 2024; Csordás et al., 2024; Zhu et al., 2024; Gu & Dao, 2023b; De et al., 2024; Beck et al., 2024).

To this end, our work squarely focuses on understanding the representation power of Mamba, and in particular its ICL capability, which, while extensively studied for transformers (Xie et al., 2021; Hendel et al., 2023; Bai et al., 2023), remains largely unexplored for SSMs. In this space, recent studies such as Sushma et al. (2024), have shown that SSMs can perform gradient-based learning for in-context adaptation similar to transformers. There is conflicting evidence whether Mamba's ICL abilities are better (Grazzi et al., 2024) or worse (Halloran et al., 2024; Akyürek et al., 2024) compared to transformers. Nonetheless, SSMs have demonstrated promising results in in-context reinforcement learning tasks (Lu et al., 2024), as well as in next-state prediction for dynamical models (Joseph et al., 2024), highlighting the potential of SSMs as efficient alternatives to transformers for ICL tasks. Motivated by this, as opposed to an architectural comparison Jelassi et al. (2024); Bhattamishra et al. (2024); Merrill et al. (2024); Sarrof et al. (2024), here we solely focus on Mamba's ICL capabilities, specifically, through the lens of random Markov processes. This framework has been successfully applied to transformers (Edelman et al., 2024; Makkuva et al., 2025; 2024; Rajaraman et al., 2024; Nichani et al., 2024), where it helped unveil fundamental learning mechanisms of transformers such as induction heads. Ours is the first work that employs this framework for Mamba and SSMs.

## 2 PROBLEM SETUP

We formally define the problem setting and provide necessary background. We use the following notation: scalars are denoted by such italic lower case letters as $x, y$, Euclidean vectors by bold $\boldsymbol{x}, \boldsymbol{y}$, and matrices by upper case $X, Y$, etc. $\mathbf{1}$ refers to the all-one vector. For $T \in \mathbb{N}$, $[T] \triangleq \{1, \ldots, T\}$, and for a sequence $(x_t)_{t \geq 1}$, define $x_k^t \triangleq (x_k, \ldots, x_t)$. For $z \in \mathbb{R}$, $\mathrm{sigmoid}(z) \triangleq 1/(1 + e^{-z})$, $\mathrm{ReLU}(z) \triangleq \max(0, z)$, and $\mathrm{softplus}(z) \triangleq \log(1 + e^z)$. $\mathrm{Unif}(S)$ denotes the uniform distribution over a set $S$ and $\mathrm{Dir}(\boldsymbol{\beta})$ denotes the Dirichlet distribution with parameter $\boldsymbol{\beta} > 0$. $D_{\mathrm{KL}}(P \| Q)$ denotes the KL divergence between distributions $P$ and $Q$.

### 2.1 INPUT DATA: RANDOM MARKOV CHAINS

To investigate the ICL capabilities of Mamba, we build upon the Markov-ICL framework of Edelman et al. (2024). In particular, we let the input tokens to be stochastic and drawn from a random Markov chain of order $k$. That is, the token sequence $x = (x_t)_{t=1}^T \in \mathcal{X}^T$ on the state space (vocabulary) $\mathcal{X}$ follows the transition dynamics:

$$\mathbb{P}\left(x_{t+1} = \cdot \mid x_1^t\right) = \mathbb{P}\left(x_{t+1} = \cdot \mid x_{t-k+1}^t\right), \tag{1}$$

almost surely for all $t \in [T]$, and the $k^{\text{th}}$-order Markov kernels, $\mathbb{P}\left(x_{t+1} = \cdot \mid x_{t-k+1}^t = i_{t-k+1}^t\right)$, are sampled independently for each tuple $(i_{t-k+1}, \cdots, i_t)$ from the Dirichlet prior $\mathrm{Dir}(\beta \cdot \mathbf{1})$, with $\beta > 0$. When $\beta = 1$, this corresponds to the uniform distribution on the $S$-dimensional simplex $\Delta_1^S$, where size $S = |\mathcal{X}|$.

The transition matrix $P = (P_{i_1^k})_{i_1^k \in \mathcal{X}^k}, P_{i_1^k} \in [0, 1]^S$, encapsulates the set of all $S^k$ conditional probabilities of the chain, each row corresponding to one of them. While this transition matrix governs the generation of each token $x_t$ for $t > k$, the first $k$-tokens $x_1, \ldots, x_k$ are drawn i.i.d. from $\mathrm{Unif}(\mathcal{X})$. This constitutes the joint law of the random variables $(P, x)$, termed random Markov distribution henceforth. More succinctly,

**Data generation** (Random Markov sequences).

1. Draw $P$ with each row sampled i.i.d. from $\mathrm{Dir}(\beta \cdot \mathbf{1})$.
2. For $t = 1, \ldots, k$, sample $x_t \sim \mathrm{Unif}(\mathcal{X})$.
3. For $t = k, \ldots, T$, sample $x_{t+1} \sim P_{x_{t-k+1}^t}$.
4. Return the input $x = (x_t)_{t=1}^T$.
5. Repeat the above steps to generate a batch $\{x^{(b)}\}_{b \in [B]}$.

**Why Random Markov is a good testbed for ICL.** As a consequence of the generation process, every sequence follows a different Markov distribution. Therefore, at inference, a model trained on this random Markovian data has to estimate the next-token distribution in-context for every test sequence. Hence, this data class serves as a good sandbox to gauge the ICL capabilities of Mamba, which was also used in a similar context for transformers (Nichani et al., 2024; Rajaraman et al., 2024).

## 2.2 MAMBA ARCHITECTURE

Selective SSMs such as Mamba and Mamba-2 are a class of sequence-to-sequence models that are closely related to RNNs and classical state space models (Gu & Dao, 2023b). A key feature underpinning these models is the *selectivity mechanism*, enabling them to selectively choose inputs at every timestep, as opposed to linear time-invariant (LTI) systems. While we believe our work captures the behavior of all selective SSMs, we will specifically focus on the state-of-the-art Mamba-2 model to simplify exposition. By slight abuse of terminology, henceforth we will also refer to this model simply as Mamba. Mathematically speaking, Mamba implements the sequence-to-sequence mapping $\mathsf{Mamba} : \mathbb{R}^{d \times T} \mapsto \mathbb{R}^{d \times T}$, where given a sequence of input embeddings $\boldsymbol{x} = (\boldsymbol{x}_t)_{t=1}^{T} \in \mathbb{R}^{d \times T}$ of dimension $d$, it outputs the corresponding output embeddings $\boldsymbol{o} = (\boldsymbol{o}_t)_{t=1}^{T} \in \mathbb{R}^{d \times T}$ of the same dimension with $\boldsymbol{o} = \mathsf{Mamba}(\boldsymbol{x})$. More precisely, fix $t \in [T]$. Then the output $\boldsymbol{o}_t$ at time $t$ is computed as $\boldsymbol{o}_t = \mathsf{Mamba}(\boldsymbol{x}_1^t)$ using the following recurrence equations (Dao & Gu, 2024):



Figure 2: Mamba-based language model.

$$H_t = a_t\,H_{t-1} + \widetilde{\boldsymbol{x}}_t\,\boldsymbol{b}_t^\top \in \mathbb{R}^{ed \times N},$$
$$\boldsymbol{y}_t = H_t\,\boldsymbol{c}_t \in \mathbb{R}^{ed},$$
$$\boldsymbol{z}_t = \boldsymbol{y}_t \odot \mathrm{ReLU}(W_z\,\boldsymbol{x}_t) \in \mathbb{R}^{ed},$$
$$\boldsymbol{o}_t = W_o\,\boldsymbol{z}_t \in \mathbb{R}^d,$$

(Mamba)

$$a_t \triangleq \exp(-a \cdot \Delta_t) \in (0,1),$$
$$\Delta_t \triangleq \mathrm{softplus}(\langle \boldsymbol{w}_\Delta, \boldsymbol{x}_t \rangle + \delta) \in \mathbb{R},$$
$$\widetilde{\boldsymbol{x}}_t \triangleq \mathrm{ReLU}(\mathrm{conv}_X(W_X\,\boldsymbol{x}_{t-w+1}^t)) \cdot \Delta_t,$$
$$\boldsymbol{b}_t \triangleq \mathrm{ReLU}(\mathrm{conv}_B(W_B\,\boldsymbol{x}_{t-w+1}^t)),$$
$$\boldsymbol{c}_t \triangleq \mathrm{ReLU}(\mathrm{conv}_C(W_C\,\boldsymbol{x}_{t-w+1}^t)),$$

(Input selectivity)

where the initial state $H_0 = 0$, $W_z \in \mathbb{R}^{ed \times d}, W_o \in \mathbb{R}^{d \times ed}, a \geq 0, \boldsymbol{w}_\Delta \in \mathbb{R}^d, \delta \in \mathbb{R}, W_X \in \mathbb{R}^{ed \times d}, W_B \in \mathbb{R}^{N \times d}$ and $W_C \in \mathbb{R}^{N \times d}$ are all learnable parameters, and $\mathrm{conv}(\boldsymbol{z}_{t-w+1}^t)$ is a time-wise convolution of window $w \in \mathbb{N}$ with distinct kernels per dimension. Here $e \in \mathbb{N}$ is the feature expansion factor, typically 2. Let $\boldsymbol{\theta}_{\mathsf{Mamba}}$ denote the set of all these parameters.

**Intuition behind Mamba.** The underlying intuition behind the update equations in Mamba is simple: given a sequence of input embeddings $(\boldsymbol{x}_t)$, we first capture their local temporal information using separate convolutions to compute $\widetilde{\boldsymbol{x}}_t$, $\boldsymbol{b}_t$, and $\boldsymbol{c}_t$ (Input selectivity). Equipped with this local memory, we perform a linear state update to compute the current state $H_t$ from the past $H_{t-1}$, weighed by an input-dependent decay factor $a_t \in (0,1)$, and $(\widetilde{\boldsymbol{x}}_t, \boldsymbol{b}_t)$. Subsequently, we compute the state projection $\boldsymbol{y}_t$, modulate it with an input-selective term to yield $\boldsymbol{z}_t$, and finally project it down to get the output embedding $\boldsymbol{o}_t$, which is a function of the entire input sequence until then, $\boldsymbol{x}_1^t$, i.e., $\boldsymbol{o}_t = \mathsf{Mamba}(\boldsymbol{x}_1^t)$.

**Mamba-based language model.** Mamba block is then incorporated into a full-fledged language model as follows:

$$x_t \in \{0,1\} \xrightarrow{\text{Embedding}} \boldsymbol{x}_t \xrightarrow{\text{Mamba}} \boldsymbol{u}_t \xrightarrow{\text{MLP}} \boldsymbol{v}_t \xrightarrow{\text{Linear}} \mathrm{logit}_t \xrightarrow{\text{Prediction}} f_{\boldsymbol{\theta}}(x_1^t), \qquad (2)$$

where $f_{\boldsymbol{\theta}}(x_1^t) \triangleq \mathbb{P}_{\boldsymbol{\theta}}(x_{t+1} = \cdot \mid x_1^t) = \mathrm{softmax}(\mathrm{logit}_t) \in [0,1]^S$ is the probability estimation for the next symbol $x_{t+1}$ conditioned on the past $x_1^t$. We omit the layer norm here for simplicity. We compactly denote the set of all model parameters as $\boldsymbol{\theta} \in \mathbb{R}^D$. We refer to § B for more details.

## 2.3 LEARNING TASK: NEXT-TOKEN PREDICTION

With the objective of auto-regressively estimating the next token, we train the model parameters $\boldsymbol{\theta}$ to minimize the cross-entropy loss between the next-token predicted probability $f_{\boldsymbol{\theta}}(x_1^t)$ and the corresponding ground-truth symbol $x_{t+1}$ across all the positions $t \in [T]$:

$$L(\boldsymbol{\theta}) \triangleq -\frac{1}{T} \sum_{t \in [T]} \mathbb{E}_P \mathbb{E}_{x_1^{t+1} \sim P} \big[ \log f_{\boldsymbol{\theta}}^{(x_{t+1})}(x_1^t) \big], \tag{3}$$

where $f_{\boldsymbol{\theta}}^{(j)}(x_1^t) \triangleq \mathbb{P}_{\boldsymbol{\theta}}(x_{t+1} = j \mid x_1^t)$ for $j \in \mathcal{X}$, and the expectation is both over the transition kernels $P$ and the Markov sequences $x = (x_t)_{t=1}^T$ sampled from $P$. In practice, it is replaced by empirical average across a finite set of batches, sampled according to the random Markov distribution in Sec. 2.1. For our experiments we use the AdamW optimizer (Kingma & Ba, 2015).

## 2.4 OPTIMAL ESTIMATOR: LAPLACIAN SMOOTHING

Given the Bayesian prediction loss in Eq. (3), it is natural to ask: *what is the optimal $\boldsymbol{\theta}$ minimizing it?* It follows from a classical result in statistics (Rissanen (1984), § A) that this minimum is achieved when the corresponding model prediction matches the (average) ground-truth predictive distribution, i.e. $\mathbb{P}_{\boldsymbol{\theta}}(x_{t+1} = j \mid x_1^t) = \mathbb{E}_{P \mid x_1^t}[\mathbb{P}(x_{t+1} = j \mid x_1^t)]$, for all $t$. Given the joint distribution of the pair $(P, x_1^{t+1})$ in Sec. 2.1, where the kernel $P \sim \text{Dir}(\beta \cdot \mathbf{1})$, it can be shown (§ A) that the conditional expectation above simplifies to the well-known *Laplacian smoothing*, also known as the *add-$\beta$ estimator* (see e.g. Merhav & Feder (1998)):

$$\mathbb{P}_{\beta}^{(k)}(x_{t+1} = j \mid x_1^t) \triangleq \mathbb{E}_{P \mid x_1^t}[\mathbb{P}(x_{t+1} = j \mid x_1^t)] = \frac{n_j + \beta}{n + 2\beta}, \qquad \text{(Laplacian smoothing)}$$

where $n_j$ is the number of times token $j$ follows the current $k^{\text{th}}$-order context $x_{t-k+1}^t$ in the sequence $x_1^t$, i.e. $n_j = |\{i : (x_{i-k}^{i-1}, x_i) = (x_{t-k+1}^t, j)\}|$ and $n$ is the frequency of this context, i.e. $n = |\{i : x_{i-k}^{i-1} = x_{t-k+1}^t\}|$. Adjusting these counts by $\beta$ plays the role of additive smoothing, which avoids assigning zero probabilities to unseen events, an idea dating back to Laplace (Laplace, 1814). It is also known that the add-$\beta$ estimator is asymptotically minimax optimal, as $T \to \infty$ (Xie & Barron, 1997; Hao et al., 2018).

**How Laplacian smoothing implies ICL.** If Mamba realizes this smoothing estimator, i.e. $\mathbb{P}_{\boldsymbol{\theta}} = \mathbb{P}_{\beta}^{(k)}$, it automatically implies its ICL abilities: given a fresh test sequence at inference, in order to optimally predict the next token, it has to process the input tokens in-context to compute the relevant counts, as in the Laplacian smoothing. *But does Mamba realize this optimal counting estimator in practice?*

## 3 DOES MAMBA LEARN IN-CONTEXT ESTIMATORS?

To investigate the ICL capabilities of Mamba, we consider the problem setup described above and train Mamba and transformer models using AdamW on the next-token prediction loss in Eq. (3) on random Markov chains (we refer to § F for more experimental details). These experiments reveal interesting and rather surprising insights about Mamba:

1. Mamba learns the optimal Laplacian smoothing estimator on the Markov prediction task, even with a single layer (Fig. 1a).

2. Convolution mechanism plays a fundamental role in Mamba, more so than gating and non-linear activations, in aiding its learning abilities (Fig. 3a).

In the sequel, we expand upon these observations in detail.

**1) Mamba learns the Laplacian smoothing.** After training, we evaluate Mamba and transformers on the same test sequence fixed beforehand and compare their performance to that of the optimal Laplacian smoothing estimator. Specifically, we compare their next-token prediction probabilities with those of the add-$\beta$ estimator. Fig. 1 illustrates these results for various Markov orders, which uncovers a surprising phenomenon: *even a single-layer Mamba sharply matches the optimal*

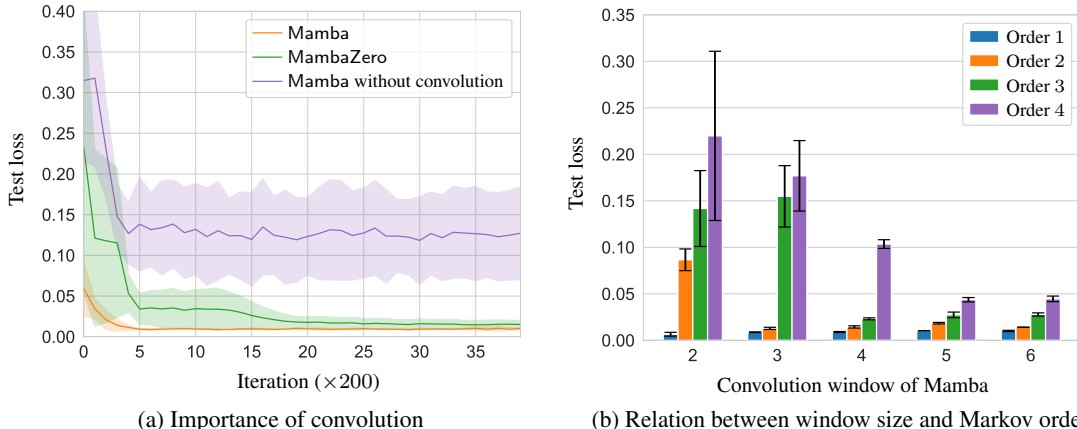

(a) Importance of convolution

(b) Relation between window size and Markov order

Figure 3: (a) illustrates the fundamental role of convolution, without which the model fails to learn the task. In contrast, a simplified variant with just the convolution (MambaZero) matches the performance of the full model. (b) highlights the relation between the Markov order $k$ and the window size $w$ of Mamba. It is required that $w \geq k + 1$ for the model to learn the order-$k$ prediction task.

*estimator on the whole sequence.* The same conclusion holds for larger state spaces and deeper models (Fig. 10a), as well as in over-parametrized settings (Fig. 9), and even when part of the dataset is held out (see § E). For transformers, we observe that a two-layer model also matches the predictor, albeit less sharply, whereas a single layer fails to solve the task. This aligns with recent theoretical results (Sanford et al., 2024; Ekbote et al., 2025), that show that two layers are required for transformers to implement an induction head (realizing the counting estimator) efficiently. We also observe that linear attention performs similarly to softmax attention in this setting (cf. Fig. 7). Sec. 4.3 further discusses Mamba vs. Transformers.

**2) Convolution is the key.** To decipher the key architectural component behind Mamba's success in Markov prediction task, we do an ablation study on its three main features: (i) convolution in Input selectivity, (ii) ReLU non-linearity in Input selectivity, and (iii) the gating mechanism in Mamba and MLP. Amongst them, interestingly, *convolution* plays a fundamental role in the model's performance, as illustrated in Fig. 3a. Here we compare the full Mamba architecture from Sec. 2.2, Mamba with just the convolution in Input selectivity removed, and a simplified Mamba architecture with only convolution (MambaZero in Sec. 4.1). Further experiments on adding/removing convolution to Mamba and transformers, as well as experiments with varying width, are shown in § E.6, while experiments on natural language are deferred to Sec. 5.2. As a metric of comparison, we use the closeness of each of these models' losses $L(\boldsymbol{\theta})$ to that of the optimal add-$\beta$ estimator $L_\beta$, i.e. i.e. $|L(\boldsymbol{\theta}) - L_\beta|$. The closer this metric is to zero, the better the model's performance is. Remarkably, the simplified Mamba with just the convolution succeeds on the Markov prediction task, while the full model without convolution fails, highlighting its fundamental importance. This raises a natural question: *how does convolution help Mamba to implement the optimal Laplacian estimator?*

## 4 HOW MAMBA IMPLEMENTS THE LAPLACIAN ESTIMATOR

Motivated by its success in learning the optimal estimator, here we study how Mamba represents Laplacian smoothing. Specifically, we provide a concrete theoretical construction backed by empirical results, that illustrates the mechanism Mamba uses to implement the estimator in practice.

### 4.1 MAMBAZERO: SIMPLIFIED MODEL

Building upon the insight that Mamba with just the convolution achieves the same performance as that of the full model (Fig. 3a), we consider its simplified version: MambaZero. MambaZero retains only the essential elements of the full model in Sec. 2.2: the Embedding layer, the convolution

inside the Mamba block in Input selectivity, and the Linear layer. More formally, it is given by:

$$
\begin{aligned}
\boldsymbol{x}_t &= \boldsymbol{e}_{x_t} \in \mathbb{R}^d, && \text{(Embedding)} \\
\boldsymbol{u}_t &= \boldsymbol{x}_t + \mathsf{MambaZero}(\boldsymbol{x}_1^t), && \text{(MambaZero)} \\
\mathrm{logit}_t &= W_\ell\, \boldsymbol{u}_t \in \mathbb{R}^S, && \text{(Linear)} \\
f_{\boldsymbol{\theta}}(x_1^t) &= \left(\mathrm{logit}_t / \|\mathrm{logit}_t\|_1\right), && \text{(Prediction)}
\end{aligned}
$$

$$
\begin{aligned}
H_t &= a_t H_{t-1} + \widetilde{\boldsymbol{x}}_t\, \boldsymbol{b}_t^\top \in \mathbb{R}^{ed \times N}, \\
\boldsymbol{y}_t &= H_t\, \boldsymbol{c}_t \in \mathbb{R}^{ed}, && \text{(MambaZero)} \\
\boldsymbol{o}_t &= W_o\, \boldsymbol{y}_t \in \mathbb{R}^d,
\end{aligned}
$$

where $\boldsymbol{e}_x$ is the token embedding for $x \in \mathcal{X}$, and the input-selective terms $a_t, \widetilde{\boldsymbol{x}}_t, \boldsymbol{b}_t$ and $\boldsymbol{c}_t$ are computed as in Input selectivity without ReLU and just the convolution. Here we use the $L_1$ normalization instead of the $\mathrm{softmax}$ in the Prediction layer to ease theoretical analysis, similar to Nichani et al. (2024); Rajaraman et al. (2024). Let $\boldsymbol{\theta} = (\{\boldsymbol{e}_i\}_{i \in \mathcal{X}}, \boldsymbol{\theta}_{\mathsf{MambaZero}}, W_\ell) \in \mathbb{R}^D$ denote the full set of parameters for appropriate $D \geq 1$.

## 4.2 MAIN THEOREM: MAMBA REPRESENTS THE LAPLACIAN ESTIMATOR

We now present our main theorem that MambaZero can represent Laplacian smoothing for any finite-state first-order Markov process. A key defining feature of our constructive proof is that it aligns with the structures empirically learned by the model, shedding light on the fundamental learning mechanisms of Mamba.

**Theorem 1.** *For a state space $\mathcal{X} = \{1, 2, \ldots, S\}$ of size $|\mathcal{X}| = S$, there is a choice of parameters for the canonical* MambaZero *model, with dimensions $N = S$, $d = 2S$, $e = 1$ and convolution window $w = 2$, such that its output prediction exactly matches that of the Laplacian estimator, for first-order Markov chains on $\mathcal{X}$. More formally, for any $\beta > 0$, there exists a set of parameters $\boldsymbol{\theta}$ such that, for all sequences $(x_t)_{t \geq 1}$ and all $t \geq 1$,*

$$
D_{\mathrm{KL}}\left(\mathbb{P}_\beta^{(1)}(\cdot \mid x_1^t) \| \mathbb{P}_{\boldsymbol{\theta}}\left(\cdot \mid x_1^t\right)\right) = 0.
$$

**Remark.** The KL divergence above is precisely the penalty paid in the cross-entropy loss in Eq. (3) at time $t$ when using the predictor $\mathbb{P}_{\boldsymbol{\theta}}$ instead of the optimal $\mathbb{P}_\beta^{(1)}$. In other words, the result implies that the loss of MambaZero can be made exactly equal to the optimal.

### 4.2.1 KEY MECHANISM AND PROOF SKETCH

**Main idea.** To build our intuition towards how MambaZero can realize the add-$\beta$ counting estimator for first-order Markov sequences, let's focus on the core MambaZero block. The key observation here is the following: if the state $H_{t-1}$ can capture all the transition counts $i \to j$ till $x_1^{t-1}$, the new state $H_t$ can be updated to account for the current transition $x_{t-1} \to x_t$ on top of the existing counts, by a suitable choice of $a_t, \widetilde{\boldsymbol{x}}_t$, and $\boldsymbol{b}_t$. Then the relevant count information corresponding to the current prefix $x_t$ could be read off from the state projection $\boldsymbol{y}_t = H_t \boldsymbol{c}_t$, and be modified to account for $\beta$-smoothing via the Linear and Prediction layers. Buttressing this idea are two key empirical facts, which in fact hold for any $k \geq 1$, underpinning our construction:

**(i) State-to-state transition factor $a_t \approx 1$ for all $t \geq 1$.** We empirically observe that when the MambaZero model is trained on random first-order Markov data, at convergence we have $a_t \approx 1$ for all $t \geq 1$ (Fig. 4). Since $a_t$ modulates how much past information flows into the present, $a_t = 1$ is required for the state $H_t$ to store all previous transition counts. Note that this can be easily achieved by setting either $a$ or $\Delta_t$ to be zero in Input selectivity, which we empirically observe as well.

**(ii) Convolution window $w \geq k + 1$.** Recalling that $k$ is the Markov order, we empirically observe that the window size $w = k + 1$ is sufficient for the full Mamba to learn the Laplacian smoothing on $k^{\text{th}}$-order Markov chains (Fig. 3b). To understand why, note that in the MambaZero architecture above, apart from the MambaZero block, all remaining equations operate on the current token at time $t$. In the MambaZero block, the dependency of the output $\boldsymbol{y}_t$ on the previous tokens is due to that of the state $H_t$ on $(\widetilde{\boldsymbol{x}}_t, \boldsymbol{b}_t)$ in the update equation, and of $\boldsymbol{c}_t$ in the state projection. Since $(\widetilde{\boldsymbol{x}}_t, \boldsymbol{b}_t, \boldsymbol{c}_t)$ depend on the past through the convolutions, a window of size $k + 1$ enables them to keep track of the current token as well as its length-$k$ prefix, which is necessary to compute the counts needed in Laplacian smoothing. On the other hand, if $w_X, w_B \leq k$, then one can find *confusable* sequences, i.e. sequences that share the same number of occurrences of all length-$k$ prefixes, but whose counts

of the tokens following each prefix is different, resulting in the model's estimate to deviate from that of the optimal add-$\beta$. We refer to § C.1 for more details. While having all the window sizes $w_X, w_B, w_c \geq k+1$ is sufficient, it can be further strengthened to $w_c = k$ (§ C.1).

We now detail our construction for the first-order case, capitalizing on these insights.

**Construction.** Let us fix $w = k + 1 = 2$. Then, $\widetilde{x}_t$ and $b_t$ only depend on the current token $x_t$ and the previous one $x_{t-1}$, while $c_t$ only depends on $x_t$. Thus, $\widetilde{x}_t$ and $b_t$ can only take $S^2$ possible values depending on the last transition in the sequence, whereas $c_t$ only $S$. To ease the notation, we will denote these values by $\widetilde{x}^{(ij)}, b^{(ij)}$, and $c^{(i)}$ respectively, for $i, j \in \mathcal{X}$. Additionally, at $t = 1$, these terms depend only on the current symbol, taking two additional values each, denoted by $\widetilde{x}^{(i)}, b^{(i)}$. Let $n_{ij}$ denote the number of transitions $i \to j$ in the input sequence $x_1^t$. Then, unfolding the state update recursion in MambaZero, we get that the output of the MambaZero block is

$$o_t = W_o\, \widetilde{x}_0 b_0^\top c_t + \sum_{ij} n_{ij}\, W_o\, \widetilde{x}^{(ij)} b^{(ij)\top} c_t. \tag{4}$$

While the output in Eq. (4) depends on all the transition counts, in view of Laplacian smoothing, we ideally want only those counts pertaining to relevant transitions, i.e. if $x_t = 0$, the counts $n_{0j}$, for $j \in \mathcal{X}$, and similarly for other values of $x_t$. To this end, we empirically observe that at convergence, the model's parameters are such that $b^{(ij)\top} c_t \approx 0$ whether $i \neq x_t$ (cf. Fig. 5). Due to this property, only the counts that are involved in the computation of the Laplacian estimator for the current token $x_t$ appear in the output $o_t$. Stitching these facts, the final logits in the Linear layer depend on the first and current token via

$$\text{logit}_t = W_\ell\, x_t + W_\ell W_o\, \widetilde{x}_0 b_0^\top c_t + \sum_j n_{x_t,j} W_\ell W_o\, \widetilde{x}^{(x_t,j)} b^{(x_t,j)\top} c_t. \tag{5}$$

The final step is to then show that for properly chosen parameters, one can make the two vectors associated with the counts to be orthogonal, and the other vectors, independent of the counts, to sum up to the vector $\beta \mathbf{1}$ (which we also empirically verified, cf. Fig. 5). Subsequently, the $L_1$ normalization in Prediction layer will give a next-token probability estimate, matching that of the add-$\beta$ estimator. We defer the full proof and additional details to § C.

**Dimension reduction for binary state space.** Interestingly, for the binary case $\mathcal{X} = \{0, 1\}$, it is possible to further reduce the hidden dimension $d = 2S$ in Thm. 1 to $d = S = 2$ by leveraging the relationship between the transition counts. The key theoretical insight is that the transition counts in binary sequences are strongly correlated. Specifically, $n_{01}$ and $n_{10}$ are at most one apart: every time a transition $0 \to 1$ occurs, either the sequence is followed by only 1's until the end, or a subsequent transition $1 \to 0$ also occurs. Therefore, the dependency of the output on $n_{01}$ is in fact a dependency on $n_{10}$. One can leverage this property to help MambaZero realize Laplacian smoothing with just two-dimensional embeddings, with arbitrarily small error. We refer to Thm. 4 in § C.3 for full details.

### 4.3 Lower bound: fundamental limit on the representation power of Mamba

We now provide a fundamental limit on the representation power of recurrent architectures like Mamba, in the form of a lower bound on the hidden dimension that is required to represent the optimal estimator. In particular, our result establishes that with finite bit precision, irrespective of depth, for any recurrent architecture to implement the Laplacian estimator, the hidden dimension has to at least scale as $\Omega(2^k)$.

**Theorem 2.** *Consider a recurrent model of the form*

$$H_t = h_t(H_{t-1}, x_t),$$
$$y_t = \mathbb{P}_{\boldsymbol{\theta}}\left(\cdot \mid x_1^t\right) = g_t(H_t),$$

*with transformations $(h_t, g_t)$, where $H_t \in \mathbb{R}^d$ and the model has a bit precision of $\mathrm{p}$. Suppose that the $k^{th}$-order Markov kernel $P$ is sampled from the Dirichlet prior with $\beta = 1$, $P \sim \mathrm{Dir}(1 \cdot \mathbf{1})$. Suppose also that the recurrent architecture satisfies the following point-wise guarantee: for any sufficiently large $t$, almost surely over $P$ and $x_1^t \sim P$,*

$$\left\| \mathbb{P}_{\boldsymbol{\theta}}\left(\cdot \mid x_1^t\right) - \mathbb{P}_1^{(k)}(\cdot \mid x_1^t) \right\|_\infty \leq \varepsilon, \tag{6}$$

where $\mathbb{P}_1^{(k)}(\cdot \mid x_1^t)$ is the Laplacian estimator for $\beta = 1$. Then, the recurrent architecture must satisfy

$$d \cdot \mathtt{p} \geq 2^k (1 - 3\varepsilon) \log(1/\varepsilon).$$

We defer the full proof and additional details to App. D.

**Depth.** We note that Thm. 2 does not assume depth one, and holds for recurrent models of any depth.

**Mamba vs. Transformers.** As Thm. 2 demonstrates, to capture a $k^{\text{th}}$-order Markov process, Mamba requires the hidden dimension to scale exponentially in $k$, whereas the best known result for transformers needs a three layer model with the hidden dimension growing linearly in $k$ (Rajaraman et al., 2024). On the other hand, for first-order sources we empirically observe from Fig. 1a that 1-layer Mamba tracks the optimal estimator more sharply than a transformer (see § E for additional comparative results). While these comparisons are meant to provide a more detailed context for Mamba, we would like to emphasize that the main focus of our paper is not a comparative study but rather a *fundamental understanding of Mamba's ICL abilities*.

**Higher orders and learning dynamics.** While Thm. 1 demonstrates that Mamba can represent the optimal estimator for finite-state first-order processes, our empirical results in Fig. 1b strongly suggest that a similar conclusion holds for higher-order sources. In a similar vein, analyzing Mamba's learning dynamics in its convergence to this smoothing estimator is an interesting topic of future research, but outside the scope of this paper, whose focus is on representation power.

## 5 BEYOND MARKOV

### 5.1 SWITCHING MARKOV MODEL

A key component of Mamba enabling selectivity is the state-transition factor $a_t$, that controls the flow of information from the past state $H_{t-1}$ to the current $H_t$: if $a_t = 1$, the past information is fully utilized in computing the current state, and hence the output, whereas $a_t = 0$ completely ignores the past. In the Markovian setting considered so far, the role of $a_t$ has largely been dormant: $a_t \approx 1$ for all $t \geq 1$, as the optimal Laplacian predictor requires counts of all transitions, demanding the use of full past (Sec. 4.2). To better highlight this selectivity mechanism, we consider a non-Markovian process, where the role of $a_t$ becomes fundamental.

Specifically, we focus on the *switching Markov process*, where we add a *switch* token to the binary alphabet, i.e. we consider $\mathcal{X} = \{0, 1, \mathrm{S}\}$. The key difference here compared to the random Markov generation in Sec. 2.1 is that until we hit switch token, we follow the same binary Markov sequence generation as the former, but once the switch state is reached, we sample a new Markov kernel and then generate a new Markov sequence. The switch tokens are sampled according to a parallel i.i.d. Bernoulli process with probability $p_{\text{switch}}$ (0.01 in our experiments). The sampling process is described in detail in § E.7. With this data model, the optimal prediction strategy is to use the add-$\beta$ estimator in between two switch tokens, and reset the transition counts every time a switch occurs. We provide empirical evidence in § E.7. Indeed, Fig. 13 illustrates that Mamba implements precisely this strategy, closely tracking the switching events via the transition factor $a_t$: it sets $a_t$ to be zero whenever $x_t = \mathrm{S}$ and to one otherwise.

### 5.2 NATURAL LANGUAGE MODELING

To test the generality of our finding that convolution plays a key role on Markovian data (Fig. 3), we conduct experiments on language modeling using the WikiText-103 dataset. Details on the experimental setup can be found in § F. By adding or removing convolution in both these models, we obtain the results in Table 1. The results illustrate that convolution enhances the performance of the two architectures, in particular for Mamba (11% vs. 2%),

Table 1: Perplexity results on the WikiText-103 dataset.

| Model | Params. | Perplexity |
|---|---|---|
| Mamba-2 (w/o conv) | 14.53 M | 30.68 |
| Mamba-2 (w/ conv) | 14.54 M | **27.55** |
| Transformer (w/o conv) | 14.46 M | 29.28 |
| Transformer (w/ conv) | 14.46 M | **28.67** |

highlighting its saliency. Further ablation studies on this task show that together with convolution, gating also plays a central role (17% change, cf. § E.8, Table 4). Furthermore, additional experiments with deeper models show that the relative importance of convolution seems to decrease as the number of layers increases (cf. § E.8, Table 5). This may be due to the fact that the role of convolution is taken over by other Mamba layers, which are known to successfully approximate convolution (Wang & Xue, 2023).

## 6 CONCLUSION

Structured state space sequence models (SSMs) and Selective SSMs such as Mamba have shown remarkable inference speed-ups over transformers while achieving comparable or superior performance on complex language modeling tasks. In this paper, we studied in-context learning (ICL) capabilities of Mamba on random Markov chains and show that, unlike transformers, even a single-layer Mamba efficiently learns the in-context Laplacian smoothing estimator. To explain this, we theoretically and empirically characterized the representation capacity of Mamba, which revealed the fundamental role of convolution, together with selectivity and recurrence, in enabling it. We further provided additional empirical results on non-Markovian data, showing the generality of our insights. Extending our results to deeper Mamba models, as well as investigating Mamba's learning dynamics, are some interesting future directions.

### ACKNOWLEDGMENTS

The work in this manuscript was partially supported by the Swiss National Science Foundation under Grant 200364.

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

## A PRELIMINARIES ON LAPLACIAN SMOOTHING

Laplacian smoothing is a mature and well understood topic. An account can be found, e.g., in Merhav & Feder (1998); Cesa-Bianchi & Lugosi (2006), with some recent updates in Bondaschi & Gastpar (2024; 2025). For the sake of completeness, we provide a brief outline of how it applies to our context. For $k$-th order Markov data, at every time instant $t$, the Laplacian add-$\beta$ estimator applied to the subsequence of tokens with the same context $i_1^k \in \mathcal{X}^k$ as the current one is the predictor that minimizes the Bayesian cross-entropy loss in Eq. (3), when the Markov kernel is sampled according to the product Dirichlet distribution $\mathrm{Dir}(\beta \cdot \mathbf{1})$. We first give an intuition of why this is the case, and we provide a full proof at the end of the section. We consider the binary case $\mathcal{X} = \{0, 1\}$, but the results can be extended to arbitrary finite alphabets.

Consider a given sequence $(x_t)_{t=1}^T$. For every length-$k$ context $i_1^k \in \mathcal{X}^k$, let $(x_t)|_{i_1^k}$ be the subsequence of tokens preceded by $i_1^k$. Note that, since each sequence $(x_t)$ is generated by a $k$-th order Markov chain, all the tokens in the sequence with the same length-$k$ prefix share the same conditional probability distribution. Furthermore, since each of the conditional distributions of the chain is randomly chosen independently from the others, the subsequence $(x_t)|_{i_1^k}$ is a sufficient statistic to estimate the probability distribution of all the tokens with the same prefix $i_1^k$. Therefore, the optimal prediction for a sequence $(x_t)_{t=1}^T$ is given by employing the optimal predictor for each i.i.d. subsequence $(x_t)|_{i_1^k}$, for every $i_1^k \in \mathcal{X}^k$. Since each conditional distribution is sampled from a Dirichlet distribution with parameter $\beta$, it is well known that the optimal predictor for such subsequences is the add-constant estimator, with constant equal to $\beta$. More specifically, if $x_{t-k}^{t-1} = i_1^k$, then the optimal estimation for $x_t$ is

$$\mathbb{P}_\beta^{(k)}\left(x_{t+1} = j \mid x_1^t\right) = \frac{n_j + \beta}{n + 2\beta}, \tag{7}$$

where $n_j$ is the number of times token $j$ appears in the subsequence $x_1^t|_{i_1^k} = (x_\ell \in x_1^t : x_{\ell-k}^{\ell-1} = i_1^k)$, and $n$ is the length of the subsequence.

We now provide a formal proof of this fact.

**Theorem 3.** *Consider the class of all $k$-th order Markov kernels $P = (P_{i_1^k})_{i_1^k \in \mathcal{X}^k}$, where each $P_{i_1^k} = \mathbb{P}(\cdot \mid i_1^k)$ is a probability distribution on $\mathcal{X} = \{0, 1\}$. Let each $P_{i_1^k}$ be sampled i.i.d. from $\mathrm{Dir}(\beta \cdot \mathbf{1})$, and let $x_1^k \sim \mathrm{Unif}(\mathcal{X}^k)$ and $x_{t+1}|x_1^t \sim P_{x_{t-k+1}^t}$. Then, the predictor $f^{(j)}(x_1^t) = \hat{\mathbb{P}}(x_{t+1} = j \mid x_1^t)$, for $j \in \{0, 1\}$, that minimizes the loss*

$$L \triangleq -\frac{1}{T} \sum_{t \in [T]} \mathbb{E}_P \mathbb{E}_{x_1^{t+1} \sim P}\left[x_{t+1} \cdot \log f^{(1)}(x_1^t) + (1 - x_{t+1}) \cdot \log f^{(0)}(x_1^t)\right] \tag{8}$$

*is the add-$\beta$ estimator in Eq. (7), i.e. the minimizer $f_*^{(j)}(x_1^t) = \mathbb{P}_\beta^{(k)}(x_{t+1} = j \mid x_1^t)$, for all $t \geq k$.*

*Proof.* First note that

$$L = -\frac{1}{T} \sum_t \mathbb{E}_P \mathbb{E}_{x_1^{t+1} \sim P}\left[x_{t+1} \cdot \log f^{(1)}(x_1^t) + (1 - x_{t+1}) \cdot \log f^{(0)}(x_1^t)\right]$$

$$= -\frac{1}{T} \sum_t \mathbb{E}_{x_1^t} \mathbb{E}_{x_{t+1}|x_1^t}\left[x_{t+1} \cdot \log f^{(1)}(x_1^t) + (1 - x_{t+1}) \cdot \log f^{(0)}(x_1^t)\right]$$

$$= -\frac{1}{T} \sum_t \mathbb{E}_{x_1^t}\left[\mathbb{E}_{x_{t+1}|x_1^t}[x_{t+1}] \cdot \log f^{(1)}(x_1^t) + (1 - \mathbb{E}_{x_{t+1}|x_1^t}[x_{t+1}]) \cdot \log f^{(0)}(x_1^t)\right].$$

Let us define the distribution $f_*^{(1)}(x_1^t) \triangleq \mathbb{E}_{x_{t+1}|x_1^t}[x_{t+1}]$ and $f_*^{(0)}(x_1^t) \triangleq 1 - f_*^{(1)}(x_1^t)$. Then, we can rewrite the loss as

$$L = \frac{1}{T} \sum_t \mathbb{E}_{x_1^t}\left[-f_*^{(1)}(x_1^t) \cdot \log f^{(1)}(x_1^t) - f_*^{(0)}(x_1^t) \cdot \log f^{(0)}(x_1^t)\right].$$

For every $t \in [T]$ and every $x_1^t \in \mathcal{X}^t$, the term inside the expectation is minimized by picking $f^{(1)}(x_1^t) = f_*^{(1)}(x_1^t)$. In fact, note that it can be rewritten as

$$-f_*^{(1)}(x_1^t) \cdot \log f^{(1)}(x_1^t) - f_*^{(0)}(x_1^t) \cdot \log f^{(0)}(x_1^t)$$

$$= f_*^{(1)}(x_1^t) \cdot \log \frac{f_*^{(0)}(x_1^t)}{f^{(1)}(x_1^t)} + f_*^{(0)}(x_1^t) \cdot \log \frac{f_*^{(0)}(x_1^t)}{f^{(0)}(x_1^t)} - f_*^{(1)}(x_1^t) \log f_*^{(1)}(x_1^t)$$

$$- f_*^{(0)}(x_1^t) \log f_*^{(0)}(x_1^t)$$

$$= D_{\mathrm{KL}}\left(f_*(x_1^t)\|f(x_1^t)\right) + H(f_*(x_1^t)),$$

which is minimized when $D_{\mathrm{KL}}\left(f_*(x_1^t)\|f(x_1^t)\right) = 0$, i.e., when $f(x_1^t) = f_*(x_1^t)$. We will now show that $f_*(x_1^t)$ is precisely the add-$\beta$ estimator. Consider any context $i_1^k$ and any sequence $x_1^t$ such that $x_{t-k+1}^t = i_1^k$. Let also $p \triangleq P_{i_1^k}(1) = \mathbb{P}(1 \mid i_1^k)$. Then,

$$f_*^{(1)}(x_1^t) \triangleq \mathbb{E}_{x_{t+1}|x_1^t}[x_{t+1}]$$

$$= \mathbb{E}_{P_{i_1^k}|x_1^t} \mathbb{E}_{x_{t+1}|x_1^t, P_{i_1^k}}[x_{t+1}]$$

$$= \mathbb{E}_{P_{i_1^k}|x_1^t}[P_{i_1^k}(1)]$$

$$= \mathbb{E}_{P_{i_1^k}|x_1^t|_{i_1^k}}[P_{i_1^k}(1)],$$

where in the last equation we used the fact that, when $x_1^k \sim \mathrm{Unif}(\mathcal{X}^k)$, the subsequence $x_1^t|_{i_1^k}$ is a sufficient statistic for $P_{i_1^k}$. Hence,

$$f_*^{(1)}(x_1^t) = \mathbb{E}_{P_{i_1^k}|x_1^t|_{i_1^k}}[P_{i_1^k}(1)]$$

$$= \int_0^1 \frac{p^{\beta-1}(1-p)^{\beta-1}p^{n_1}(1-p)^{n_0}}{\int_0^1 q^{\beta-1}(1-q)^{\beta-1}q^{n_1}(1-q)^{n_0}\,dq} \cdot p\,dp$$

$$= \frac{\int_0^1 p^{n_1+\beta}(1-p)^{n_0+\beta-1}\,dp}{\int_0^1 q^{n_1+\beta-1}(1-q)^{n_0+\beta-1}\,dq}$$

$$= \frac{\Gamma(n_1+\beta+1)\Gamma(n_0+\beta)}{\Gamma(n+2\beta+1)} \cdot \frac{\Gamma(n+2\beta)}{\Gamma(n_1+\beta)\Gamma(n_0+\beta)}$$

$$= \frac{n_1+\beta}{n+2\beta},$$

where we used the fact that $P_{i_1^k} \sim \mathrm{Dir}(\beta \cdot \mathbf{1})$, that $\int_0^1 q^{z_1-1}(1-q)^{z_0-1} = \Gamma(z_1)\Gamma(z_0)/\Gamma(z_1+z_0)$, and that $\Gamma(z+1) = z\Gamma(z)$. $\qquad\square$

**Remark.** The proof above is for $x_1^k \sim \mathrm{Unif}(\mathcal{X}^k)$. However, note that the same proof would also work for $x_1^k$ distributed according to any distribution that is independent of the Markov kernel $P$. If instead the distribution depends on $P$ (e.g., the stationary distribution of the Markov chain), then the proof would fail in the step where $x_1^t|_{i_1^k}$ is a sufficient statistic for $P_{i_1^k}$.

**Remark.** It is important to note that, to be able to implement such a predictor requires in-context capabilities: at inference, in order to optimally predict the next token, the model must be able to look into the previous tokens of the test sequence, and count the tokens with the correct prefix.

## B MAMBA-BASED LANGUAGE MODELING ARCHITECTURE

Mamba block can be incorporated into a full-fledged language model as follows: let $x = (x_1, x_2, \cdots, x_T) \in \mathcal{X}^T$ be an input token-sequence over the alphabet $\mathcal{X}$; here $\mathcal{X} = \{0, 1\}$ as explained in Sec. 2.1. Then, at every $t \in [T]$, the output of the language model $\boldsymbol{\theta}$ is given by the following sequence of equations (Dao & Gu, 2024):

$$\boldsymbol{x}_t = \boldsymbol{e}_{x_t} \in \mathbb{R}^d, \tag{Embedding}$$

$$\boldsymbol{u}_t = \boldsymbol{x}_t + \mathsf{Mamba}(\boldsymbol{x}_1^t) \in \mathbb{R}^d, \tag{Mamba}$$

$$\boldsymbol{v}_t = \boldsymbol{u}_t + W_2[\mathrm{ReLU}(W_1 \boldsymbol{u}_t) \odot W_3 \boldsymbol{u}_t] \in \mathbb{R}^d, \tag{MLP}$$

$$\mathrm{logit}_t = W_\ell \boldsymbol{v}_t \in \mathbb{R}^S, \tag{Linear}$$

$$f_{\boldsymbol{\theta}}(x_1^t) \triangleq \mathbb{P}_{\boldsymbol{\theta}}\left(x_{t+1} = \cdot \mid x_1^t\right) = \mathrm{softmax}(\mathrm{logit}_t) \in [0, 1]^S, \tag{Prediction}$$

where the parameters $\boldsymbol{e}_i \in \mathbb{R}^d, W_1 \in \mathbb{R}^{4d \times d}, W_2 \in \mathbb{R}^{d \times 4d}$ and $W_\ell \in \mathbb{R}^{S \times d}$ are learnable, and $f_{\boldsymbol{\theta}}(x_1^t)$ is the probability law for the next symbol $x_{t+1}$ conditioned on the past $x_1^t$. We omit the layer norm here for simplicity. We compactly denote the set of all model parameters as $\boldsymbol{\theta}$, i.e. $\boldsymbol{\theta} = (\{\boldsymbol{e}_i\}_{i \in \mathcal{X}}, \boldsymbol{\theta}_{\mathsf{Mamba}}, W_{1,2,3}, W_\ell) \in \mathbb{R}^D$.

# C PRELIMINARIES AND PROOF OF THM. 1 AND THM. 4

## C.1 EMPIRICAL INSIGHTS

Here we expand upon our empirical observations in 4.2.1, which form the basis of our proof.

**State-to-state transition factor** $a_t \approx 1$ **for all** $t \geq 1$**.** We empirical evidence supporting this observation in Fig. 4.

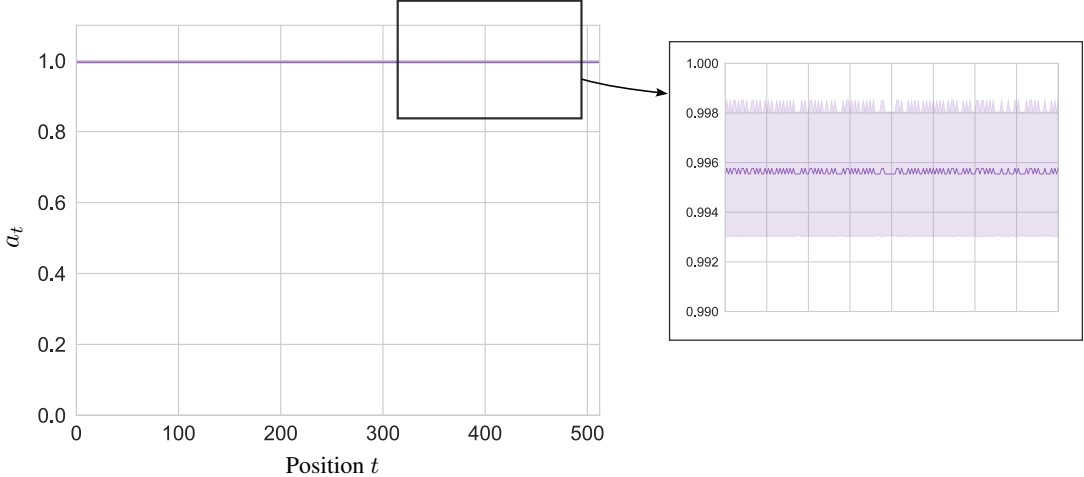

Figure 4: Value of $a_t$ across positions at convergence.

**Convolution window** $w \geq k + 1$**.** Recalling that $k$ is the Markov order, we empirically observe that the window that $w = k + 1$ is sufficient for the full Mamba to learn the Laplacian smoothing on $k^{\text{th}}$-order Markov chains. To understand why, note that in the MambaZero architecture above, apart from the MambaZero block, all remaining equations operate on the current token at time $t$. In the MambaZero block, same as the Mamba block except ReLU, the dependency of the output $\boldsymbol{y}_t$ on the previous tokens is due to that of the state $H_t$ on $(\widetilde{\boldsymbol{x}}_t, \boldsymbol{b}_t)$ in the update equation, and of $\boldsymbol{c}_t$ in the state projection. Since $(\widetilde{\boldsymbol{x}}_t, \boldsymbol{b}_t, \boldsymbol{c}_t)$ depend on the past through the convolutions, a window of size $k + 1$ enables them to keep track of the current token as well as its length-$k$ prefix, which is necessary to compute the counts needed in Laplacian smoothing. On the other hand, if $w \leq k$, then one can find *confusable* sequences, i.e. sequences that share the same number of occurrences of all length-$k$ prefixes, but whose counts of the tokens following each prefix is different.

For such sequences, the state $H_t$ is the same, and so are the predicted probabilities by the Mamba model; however, the optimal estimator, depending on the transition counts, would give very different probability estimates, allowing Mamba's prediction loss to deviate from that of the optimal. For example, consider $k = 1$. If $w = 1$, then $(\widetilde{\boldsymbol{x}}_t, \boldsymbol{b}_t, \boldsymbol{c}_t)$ depend only on the current token $x_t$. Then, consider the two sequences $x = (0, 1, 0, 1, 0, 1)$ and $\widetilde{x} = (0, 0, 0, 1, 1, 1)$. At time $t = 6$, these two sequences would give the same state $H_t$ and the same output $\boldsymbol{y}_t$, since they share the same number of tokens $0$ and $1$. Therefore, the estimated probability given by the model would be the same in both cases. However, the optimal add-constant estimator (with $\beta = 1$) would estimate the probability of $x_{t+1} = 1$ to be $1/4$ for $\boldsymbol{x}$, and $3/4$ for $\widetilde{\boldsymbol{x}}$.

Further, *it is sufficient that the convolution for* $\boldsymbol{c}_t$ *has window* $w_C = k$. That is, the convolution $\text{conv}_C$ involved in the computation of $\boldsymbol{c}_t$ can have a window size equal to the Markov order $k$ (i.e., one less than $\text{conv}_X$ and $\text{conv}_B$) without affecting the model's capability of learning the task (or, equivalently, the left-most kernel coefficients of $\text{conv}_C$ can be taken to be zero). Intuitively, this is because the role of $\boldsymbol{c}_t$ in the state projection is to select the correct transition counts for the computation of the estimator, distilled into $\boldsymbol{y}_t$. In order to do so, it is sufficient to know the length-$k$ context of the current symbol $x_t$, which can be encoded by a convolution with window size $k$.

**Orthogonal count-dependent vectors.** The inner products $\boldsymbol{b}^{(ij)\top} \boldsymbol{c}^{(k)}$ corresponding to $i \neq k$ go to zero at convergence: only the correct counts are kept in the final logit.

**Convergence to the optimal $\beta = 1$.** The count-independent part of the final logit converges to $\beta = 1$, corresponding to the optimal Laplacian estimator.

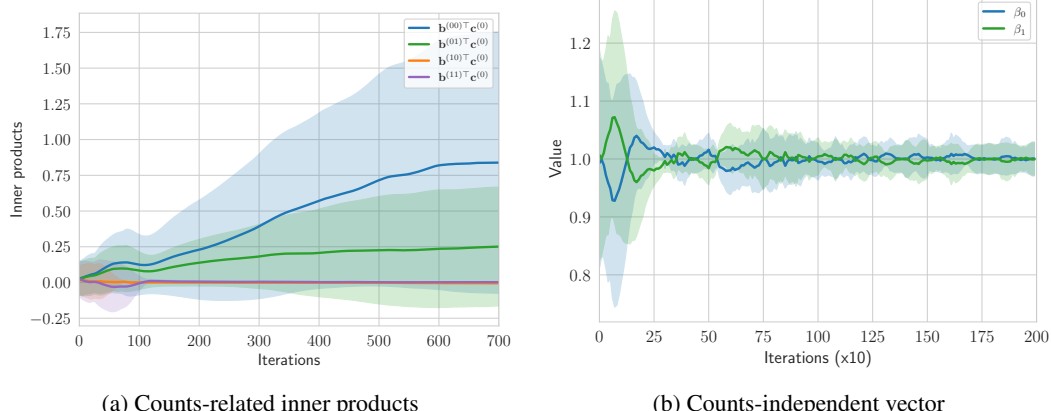

(a) Counts-related inner products

(b) Counts-independent vector

Figure 5: (a) Counts-related inner products across interations. Only the correct counts corresponding to $\boldsymbol{b}^{(ij)\top}\boldsymbol{c}^{(k)}$ for $i = k$ have a non-zero inner product at convergence. (b) Binary coordinates of the counts-independent vector across iterations. Both coordinates converge to the optimal $\beta = 1$.

## C.2 Proof of Thm. 1

Let $\beta > 0$ be the constant of the considered add-constant estimator. Let us fix $a = 0$ and $\Delta_t = 1$, so that $a_t = 1$, for all $t \geq 1$. This can be done by picking, e.g., $\boldsymbol{w}_\Delta = \boldsymbol{0}$ and $\delta$ such that $\mathrm{softplus}(\delta) = 1$. Note that the application of convolution to a given sequence of vectors $\boldsymbol{z}_1^t$ can be rewritten as a linear matrix-form operation. For example, for $\mathrm{conv}_X$, one has that

$$\mathrm{conv}_X(\boldsymbol{z}_t) = D_X^{(0)}\boldsymbol{z}_{t-1} + D_X^{(1)}\boldsymbol{z}_t \tag{9}$$

where $D_X^{(0)}$ and $D_X^{(1)}$ are diagonal matrices. The same holds for $\mathrm{conv}_B$ and $\mathrm{conv}_C$, with corresponding diagonal matrices $D_B^{(0)}, D_B^{(1)}, D_C^{(0)}$ and $D_C^{(1)}$.

Let us take the embedding vectors $\boldsymbol{e}_i, i \in \mathcal{X}$, to be the one-hot encoding vectors for the alphabet $\mathcal{X}$, interleaved with zeros, i.e., let $\boldsymbol{e}_i$ be such that $e_{i,2i-1} = 1$ and 0 otherwise. Furthermore, take $W_X$ to be the $2S \times 2S$ matrix such that $W_X(i,j) = 1$ for $i = 2k$ and $j = 2k-1$, for $1 \leq k \leq S$, and 0 otherwise. (The role of $W_X$ is shift each coordinate of the embedding vectors by one.) Take now $\mathrm{conv}_X$ to be such that its output is simply equal to the current vector, i.e., take $D_X^{(0)} = \boldsymbol{0}$ and $D_X^{(1)} = I_{2S \times 2S}$. Take also $W_B = I_{S \times S}$ and $\mathrm{conv}_B$ so that the output is equal to the second-to-last vector, i.e., take $D_B^{(0)} = I_{S \times S}$ and $D_X^{(1)} = \boldsymbol{0}$. Finally, take $W_C = I_{S \times S}$ and $\mathrm{conv}_C$ such that $C^{(0)} = \boldsymbol{0}$ and $C^{(1)} = I_{S \times S}$.

The final logit vector is in general equal to

$$\mathrm{logit}_t = W_\ell \boldsymbol{x}_t + W_\ell W_o \widetilde{\boldsymbol{x}}^{(x_1)}\boldsymbol{b}^{(x_1)\top}\boldsymbol{c}_t + \sum_{ij} n_{ij} W_\ell W_o \widetilde{\boldsymbol{x}}^{(ij)}\boldsymbol{b}^{(ij)\top}\boldsymbol{c}_t. \tag{10}$$

Using the matrices chosen above, we can simplify the formula as follows. Firstly, note that $\boldsymbol{b}^{(x_1)} = \boldsymbol{0}$, as the $\boldsymbol{b}$ vectors only depend on the second-to-last vectors. Furthermore, since the embedding vectors are orthogonal to each other, we have that $\boldsymbol{b}^{(ij)\top}\boldsymbol{c}_t = 1$ whether $i = x_t$, and 0 otherwise. (That is, only the correct counts are kept in the logit computation.) With these simplifications, the logit formula becomes

$$\mathrm{logit}_t = W_\ell \boldsymbol{x}_t + \sum_j n_{x_t,j} W_\ell W_o \widetilde{\boldsymbol{x}}^{(j)}. \tag{11}$$

Finally, take $W_o = I_{2S \times 2S}$ and take $W_\ell$ such that $W_\ell(i) = \beta \boldsymbol{1}$ for all odd $i$, $W_\ell(2i, i) = 1$ for $1 \leq i \leq S$, and 0 otherwise. With this choice, we get, for all $t \geq 1$,

$$\mathrm{logit}_t = \beta \boldsymbol{1} + \sum_j n_{x_t,j}\boldsymbol{e}_i \tag{12}$$

where $e_i$ is the one-hot vector for symbol $i \in \mathcal{X}$. After the normalization, we finally get

$$f_{\boldsymbol{\theta}}(x_1^t)_j = \frac{n_{ij} + \beta}{\sum_k n_{ik} + S\beta} \tag{13}$$

if $x_t = i$, for $i \in \mathcal{X}$. This is precisely the required add-$\beta$ Laplacian estimator.

### C.3 THM. 4: DIMENSIONALITY REDUCTION FOR THE BINARY CASE

**Theorem 4.** *For the canonical* MambaZero *model with dimensions $d = N = 2$, $e = 1$, and convolution window $w = 2$, there is a choice of parameters such that the model prediction is arbitrarily close to the Laplacian estimator for random first-order Markov chains. More formally, for any $\beta > 0$ and $\epsilon \in (0, 1)$, there exists a set of parameters $\boldsymbol{\theta}$ such that, for all sequences $(x_t)_{t\geq 1}$ and all $t \geq 1$,*

$$D_{\mathrm{KL}}\left(\mathbb{P}_\beta^{(1)}(\cdot \mid x_1^t) \| \mathbb{P}_{\boldsymbol{\theta}}\left(\cdot \mid x_1^t\right)\right) \leq \epsilon.$$

*Proof.* Fix $\epsilon > 0$ and let $\beta > 0$ be the constant of the considered add-constant estimator. Let us fix $a = 0$ and $\Delta_t = 1$, so that $a_t = 1$, for all $t \geq 1$. This can be done by picking, e.g., $\boldsymbol{w}_\Delta = \boldsymbol{0}$ and $\delta$ such that $\mathrm{softplus}(\delta) = 1$. Let us compactly denote the convolution kernels as

$$\mathrm{conv}_X = \begin{pmatrix} \alpha_{00} & \alpha_{01} \\ \alpha_{10} & \alpha_{11} \end{pmatrix}, \qquad \mathrm{conv}_B = \begin{pmatrix} \gamma_{00} & \gamma_{01} \\ \gamma_{10} & \gamma_{11} \end{pmatrix} \tag{14}$$

where each row corresponds to the kernel weights applied time-wise to each coordinate of the input sequence $(\boldsymbol{x}_t)_{t\geq 1}$. Since the window for $\mathrm{conv}_C$ is $w_C = 1$, we can simply assume w.l.o.g. that $C_t = W_C \boldsymbol{x}_t$.

Let us denote the embedding vectors to be $\boldsymbol{e}_0 = (e_{00}, e_{01})^\top$ and $\boldsymbol{e}_1 = (e_{10}, e_{11})^\top$, and assume that the vectors are not collinear. Take also $W_X = W_B$ such that

$$W_X \boldsymbol{e}_0 = \begin{pmatrix} 1 \\ 0 \end{pmatrix}, \qquad W_X \boldsymbol{e}_1 = \begin{pmatrix} 0 \\ 1 \end{pmatrix} \tag{15}$$

and take $W_C$ such that

$$W_C \boldsymbol{e}_0 = \begin{pmatrix} c_0 \\ 0 \end{pmatrix}, \qquad W_C \boldsymbol{e}_1 = \begin{pmatrix} 0 \\ c_1 \end{pmatrix}. \tag{16}$$

Let us also take the kernels of $\mathrm{conv}_X$ and $\mathrm{conv}_B$ to be the same across coordinates, i.e.,

$$\mathrm{conv}_X = \begin{pmatrix} \alpha_0 & \alpha_1 \\ \alpha_0 & \alpha_1 \end{pmatrix}, \qquad \mathrm{conv}_B = \begin{pmatrix} \gamma_0 & \gamma_1 \\ \gamma_0 & \gamma_1 \end{pmatrix} \tag{17}$$

such that the following conditions are satisfied:

$$\begin{cases} \alpha_0 \gamma_0 + \alpha_1 \gamma_1 = 0 \\ \alpha_0 \gamma_1 + \alpha_1 \gamma_0 > 0 \\ \alpha_0 \neq \alpha_1 \\ \frac{\alpha_0 \gamma_1}{\alpha_0 \gamma_1 + \alpha_1 \gamma_0} = -\beta\epsilon \end{cases} \tag{18}$$

Note that, with such a choice of parameters, we have

$$X^{(0)} = \begin{pmatrix} \alpha_1 \\ 0 \end{pmatrix}, \qquad X^{(1)} = \begin{pmatrix} 0 \\ \alpha_1 \end{pmatrix}, \qquad B^{(0)} = \begin{pmatrix} \gamma_1 \\ 0 \end{pmatrix}, \qquad B^{(1)} = \begin{pmatrix} 0 \\ \gamma_1 \end{pmatrix} \tag{19}$$

$$C^{(0)} = \begin{pmatrix} c_0 \\ 0 \end{pmatrix}, \qquad C^{(1)} = \begin{pmatrix} 0 \\ c_1 \end{pmatrix} \tag{20}$$

$$X^{(00)} = \begin{pmatrix} \alpha_0 + \alpha_1 \\ 0 \end{pmatrix}, \qquad X^{(01)} = \begin{pmatrix} \alpha_0 \\ \alpha_1 \end{pmatrix}, \qquad X^{(10)} = \begin{pmatrix} \alpha_1 \\ \alpha_0 \end{pmatrix}, \qquad X^{(11)} = \begin{pmatrix} 0 \\ \alpha_0 + \alpha_1 \end{pmatrix} \tag{21}$$

$$B^{(00)} = \begin{pmatrix} \gamma_0 + \gamma_1 \\ 0 \end{pmatrix}, \qquad B^{(01)} = \begin{pmatrix} \gamma_0 \\ \gamma_1 \end{pmatrix}, \qquad B^{(10)} = \begin{pmatrix} \gamma_1 \\ \gamma_0 \end{pmatrix}, \qquad B^{(11)} = \begin{pmatrix} 0 \\ \gamma_0 + \gamma_1 \end{pmatrix}. \tag{22}$$

(We replaced the vector notation of Sec. 4 with matrix notation, so that $X^{(0)}$ has to be intended as $\widetilde{x}^{(0)}$, and so on.) Take also $W_o = W_\ell = I$. With this choice of parameters, the final logit vector becomes, using Eq. (5),

$$\text{logit}_t = W_\ell \bigg( x_t + W_o X^{(x_1)} B^{(x_1)\top} C^{(x_t)} + \mathbb{1}_{\{x_1 \neq x_t\}} W_o X^{(x_1 x_t)} B^{(x_1 x_t)\top} C^{(x_t)}$$

$$+ n_{x_t x_t} W_o X^{(x_t x_t)} B^{(x_t x_t)\top} C_t + n_{x_t \bar{x}_t} W_o \left( X^{(x_t \bar{x}_t)} B^{(x_t \bar{x}_t)\top} + X^{(\bar{x}_t x_t)} B^{(\bar{x}_t x_t)\top} \right) C_t \bigg) \tag{23}$$

$$= \begin{pmatrix} e_{00} + c_0 \alpha_1 \gamma_1 \\ e_{01} \end{pmatrix} + \mathbb{1}_{\{x_1=1\}} \cdot \begin{pmatrix} 0 \\ c_0 \alpha_0 \gamma_1 \end{pmatrix} + n_{00} \cdot \begin{pmatrix} (\alpha_0 + \alpha_1)(\gamma_0 + \gamma_1)c_0 \\ 0 \end{pmatrix}$$

$$+ n_{01} \cdot \begin{pmatrix} (\alpha_0 \gamma_0 + \alpha_1 \gamma_1)c_0 \\ (\alpha_0 \gamma_1 + \alpha_1 \gamma_0)c_0 \end{pmatrix} \tag{24}$$

$$= \begin{pmatrix} e_{00} + c_0 \alpha_1 \gamma_1 \\ e_{01} \end{pmatrix} + \mathbb{1}_{\{x_1=1\}} \cdot \begin{pmatrix} 0 \\ c_0 \alpha_0 \gamma_1 \end{pmatrix} + n_{00} \cdot \begin{pmatrix} (\alpha_0 \gamma_1 + \alpha_1 \gamma_0)c_0 \\ 0 \end{pmatrix}$$

$$+ n_{01} \cdot \begin{pmatrix} 0 \\ (\alpha_0 \gamma_1 + \alpha_1 \gamma_0)c_0 \end{pmatrix} \tag{25}$$

if $x_t = 0$, and

$$\text{logit}_t = \begin{pmatrix} e_{10} \\ e_{11} + c_1 \alpha_1 \gamma_1 \end{pmatrix} + \mathbb{1}_{\{x_1=0\}} \cdot \begin{pmatrix} c_1 \alpha_0 \gamma_1 \\ 0 \end{pmatrix} + n_{10} \cdot \begin{pmatrix} (\alpha_0 \gamma_1 + \alpha_1 \gamma_0)c_1 \\ (\alpha_0 \gamma_0 + \alpha_1 \gamma_1)c_1 \end{pmatrix}$$

$$+ n_{11} \cdot \begin{pmatrix} 0 \\ (\alpha_0 + \alpha_1)(\gamma_0 + \gamma_1)c_1 \end{pmatrix} \tag{26}$$

$$= \begin{pmatrix} e_{10} \\ e_{11} + c_1 \alpha_1 \gamma_1 \end{pmatrix} + \mathbb{1}_{\{x_1=0\}} \cdot \begin{pmatrix} c_1 \alpha_0 \gamma_1 \\ 0 \end{pmatrix} + n_{10} \cdot \begin{pmatrix} (\alpha_0 \gamma_1 + \alpha_1 \gamma_0)c_1 \\ 0 \end{pmatrix}$$

$$+ n_{11} \cdot \begin{pmatrix} 0 \\ (\alpha_0 \gamma_1 + \alpha_1 \gamma_0)c_1 \end{pmatrix} \tag{27}$$

if $x_t = 1$. Take now

$$e_{00} = e_{11} = (\alpha_0 \gamma_1 + \alpha_1 \gamma_0)\beta c_0 - \alpha_1 \gamma_1 c_0 \tag{28}$$

$$e_{01} = e_{10} = (\alpha_0 \gamma_1 + \alpha_1 \gamma_0)\beta c_0 - \alpha_0 \gamma_1 c_0 \tag{29}$$

With this choice of parameters, after the layer normalization, the final output probability vector is

$$f_\theta(x_1^t) = \left( \frac{n_{00} + \beta}{n_{00} + n_{01} + 2\beta + \mathbb{1}_{\{x_1=0\}} \cdot \beta\epsilon}, \; \frac{n_{01} + \beta + \mathbb{1}_{\{x_1=0\}} \cdot \beta\epsilon}{n_{00} + n_{01} + 2\beta + \mathbb{1}_{\{x_1=0\}} \cdot \beta\epsilon} \right)^\top \tag{30}$$

if $x_t = 0$, and

$$f_\theta(x_1^t) = \left( \frac{n_{10} + \beta + \mathbb{1}_{\{x_1=1\}} \cdot \beta\epsilon}{n_{10} + n_{11} + 2\beta + \mathbb{1}_{\{x_1=1\}} \cdot \beta\epsilon}, \; \frac{n_{11} + \beta}{n_{10} + n_{11} + 2\beta + \mathbb{1}_{\{x_1=1\}} \cdot \beta\epsilon} \right)^\top \tag{31}$$

if $x_t = 1$. Note that the resulting predicted probabilities exactly match the add-$\beta$ estimator when $x_1 \neq x_t$, but they are slightly different when $x_1 = x_t$ due to the additional $\beta\epsilon$ factor. We now show that, when the additional factor is present, the two predictors nevertheless differ by at most $\epsilon$ in KL distance. We show it for the case $x_1 = x_t = 0$, the other case follows in the same way. In fact, note that

$$\frac{n_{01} + \beta + \beta\epsilon}{n_{00} + n_{01} + 2\beta + \beta\epsilon} = \frac{n_{01} + \beta}{n_{00} + n_{01} + 2\beta} \cdot \frac{1 + \frac{\beta\epsilon}{n_{01} + \beta}}{1 + \frac{\beta\epsilon}{n_{00} + n_{01} + 2\beta}}. \tag{32}$$

Now, since

$$1 \leq 1 + \frac{\beta\epsilon}{n_{01} + \beta} \leq 1 + \epsilon \tag{33}$$

and

$$1 \leq 1 + \frac{\beta\epsilon}{n_{00} + n_{01} + 2\beta} \leq 1 + \epsilon \tag{34}$$

we have that

$$\frac{n_{01} + \beta}{n_{00} + n_{01} + 2\beta} \leq \frac{n_{01} + \beta + \beta\epsilon}{n_{00} + n_{01} + 2\beta + \beta\epsilon} \cdot (1 + \epsilon) \tag{35}$$

but we also have

$$\frac{n_{00} + n_{01} + 2\beta + \beta\epsilon}{n_{00} + n_{01} + 2\beta} \leq 1 + \frac{\beta\epsilon}{n_{00} + n_{01} + 2\beta} \leq 1 + \epsilon, \tag{36}$$

so that

$$D_{\mathrm{KL}}\left(\mathbb{P}_\beta^{(1)}\left(\cdot \mid x_1^t\right) \| \mathbb{P}_{\boldsymbol{\theta}}\left(\cdot \mid x_1^t\right)\right) = \mathbb{P}_\beta^{(1)}\left(x_{t+1} = 0 \mid x_1^t\right) \log \frac{\mathbb{P}_\beta^{(1)}\left(x_{t+1} = 0 \mid x_1^t\right)}{\mathbb{P}_{\boldsymbol{\theta}}\left(x_{t+1} = 0 \mid x_1^t\right)}$$
$$+ \mathbb{P}_\beta^{(1)}\left(x_{t+1} = 1 \mid x_1^t\right) \log \frac{\mathbb{P}_\beta^{(1)}\left(x_{t+1} = 1 \mid x_1^t\right)}{\mathbb{P}_{\boldsymbol{\theta}}\left(x_{t+1} = 1 \mid x_1^t\right)} \tag{37}$$

$$= \frac{n_{00} + \beta}{n_{00} + n_{01} + 2\beta} \log \frac{\frac{n_{00}+\beta}{n_{00}+n_{01}+2\beta}}{\frac{n_{00}+\beta}{n_{00}+n_{01}+2\beta+\beta\epsilon}}$$
$$+ \frac{n_{01} + \beta}{n_{00} + n_{01} + 2\beta} \log \frac{\frac{n_{01}+\beta}{n_{00}+n_{01}+2\beta}}{\frac{n_{01}+\beta+\beta\epsilon}{n_{00}+n_{01}+2\beta+\beta\epsilon}} \tag{38}$$

$$\leq \frac{n_{00} + \beta}{n_{00} + n_{01} + 2\beta} \log(1 + \epsilon) + \frac{n_{01} + \beta}{n_{00} + n_{01} + 2\beta} \log(1 + \epsilon) \tag{39}$$

$$\leq \log(1 + \epsilon) \tag{40}$$
$$\leq \epsilon \tag{41}$$

concluding the proof. $\qquad \square$

## D    PROOF OF THM. 2

Consider a recurrent model of the form $H_t = h(H_{t-1}, x_t)$ and $y_t = g(H_t)$ for each $t \geq 1$ where $H_t \in \mathbb{R}^d$ and the model has a bit precision of $p$. In this proof, we will assume that the state space of the underlying Markov chain is $\{0, 1\}$. By the recurrent architecture, the predicted distribution over the next token $x_{t+k+1}$ is of the form,

$$y_{t+k} = \mathbb{P}_{\boldsymbol{\theta}} \left( x_{t+k+1} = z \mid x_1^{t+k} \right) = g(z, H_{t+k}). \tag{42}$$

Recall that the add-1 estimator is defined as,

$$\frac{n(z, x_{t+1}^{t+k}) + 1}{n(x_{t+1}^{t+k}) + 2}, \tag{43}$$

where $n(z_1^k) = \sum_{i=1}^{t+1} \mathbb{I}(x_i^{i+k-1} = z_1^k)$ indicates the number of times $z_1^k$ appears in the sequence. This is the optimal estimator for sequences drawn from the product-Dirichlet prior: for every $i_1^k$, $P(\cdot \mid i_1^k) \sim \text{Dir}(1 \cdot \mathbf{1})$, which is the distribution we will assume for this proof. Fixing $x_1^t$, we can write the add-1 estimator more explicitly as a function of $x_{t+1}^{t+k}$ as,

$$\mathbb{P}_1^{(k)} \left( x_{t+k+1} = z \mid x_1^t, x_{t+1}^{t+k} = z_1^k \right) = \frac{n(z_1^k, z) + 1}{n(z_1^k) + 2}. \tag{44}$$

Now, fixing $x_1^t$, correctness of the recurrent model means that, almost surely over $P$ drawn from the prior, and $x_1^t \sim P$ and $z_1^k \sim P(\cdot | x_1^t)$,

$$g(x_{t+k+1} = 0, H_{t+k}) \in \mathbb{P}_1^{(k)} \left( x_{t+k+1} = 0 \mid x_1^t, x_{t+1}^{t+k} = z_1^k \right) + [-\varepsilon, \varepsilon]. \tag{45}$$

where $H_{t+k}$ is a function of $x_1^t$ and $z_1^k$. As $t \to \infty$, under the randomness of the draw of $x_1^t \sim P$, by the strong law of large numbers RHS converges almost surely to the conditional distribution under $P$, almost surely over the choice of $P$ from the product-Dirichlet prior. Here we use the fact that for $P$ drawn from the product-Dirichlet prior, $P(z|z_1^k) > 0$ almost surely, and so the resulting distributions are exponentially mixing and ergodic. Namely, for each $z_1^k \in \{0, 1\}^k$, almost surely over $P$ drawn from the product-Dirichlet prior,

$$\Pr \left( \limsup_{t \to \infty} \left| \mathbb{P}_1^{(k)} \left( x_{t+k+1} = 0 \mid x_1^t, x_{t+1}^{t+k} = z_1^k \right) - P(0|z_1^k) \right| > \gamma \right) = 0 \tag{46}$$

for any $\gamma > 0$. Therefore, a necessary condition to satisfy Equation (45) is, for each $z_1^k \in \mathcal{X}^k$,

$$g(x_{t+k+1} = 0, H_{t+k}) \in P(0|z_1^k) + [-\varepsilon - \eta_P(t), \varepsilon + \eta_P(t)]. \tag{47}$$

for some $\eta_P(t)$, which is a function of $P$ satisfying $\limsup_{t \to \infty} \eta_P(t) = 0$ almost surely over $P$ drawn from the prior; note that $H_{t+k}$ is implicitly a function of $x_1^t$ and $z_1^k$. Divide the interval $[0, 1]$ into $1/\varepsilon$ disjoint intervals of size $\varepsilon$ each. Recall that $P(\cdot|z_1^k) \sim \rho = \text{Dir}(1 \cdot \mathbf{1})$, which implies that the random variable $P(0|z_1^k)$ for each fixed $z_1^k$ (randomness is over $P$) is distributed as,

$$\Pr_\rho \left[ P(0|z_1^k) = \cdot \big| z_1^k \right] = \text{Unif}([0, 1]). \tag{48}$$

Consider the buckets $B_\varepsilon = \{[0, \varepsilon], [\varepsilon, 2\varepsilon), \cdots, [1 - \varepsilon, 1]\}$. Define the function $\text{round}(p) : [0, 1] \to \{0, \cdots, |B_\varepsilon| - 1\}$ to return the index of the bucket in $B_\varepsilon$ such that $p$ falls in that bucket.

**Lemma 1.** *Consider any function $f(z_1^k) : \mathcal{X}^k \to \{0, \cdots, |B_\varepsilon| - 1\}$ such that, pointwise,*

$$|\text{round}(P(0|z_1^k)) - f(z_1^k)| \leq r. \tag{49}$$

*Then, when $P(0|z_1^k) \overset{i.i.d.}{\sim} \text{Dir}(1 \cdot \mathbf{1})$,*

$$H_{Shannon}(\{f(z_1^k) : z_1^k \in \{0, 1\}^k\}) \geq 2^k \left((1 - 3\varepsilon) \log(1/\varepsilon) - \log(2r + 1)\right) \tag{50}$$

*where the randomness is over the draw of $P$ and $H_{Shannon}$ is the discrete Shannon entropy.*

*Proof.* Recall that $P(0|z_1^k) \overset{\text{i.i.d.}}{\sim} \text{Unif}([0,1])$ across $z_1^k \in \mathcal{X}^k$. Then,

$$\Pr(\text{round}(P(0|z_1^k)) = j) = \Pr(P(0|z_1^k) \in [\varepsilon(j-1/2), \varepsilon(j+1/2))) \tag{51}$$

$$= \begin{cases} \varepsilon & \text{if } 1 \le j \le |B_\varepsilon| - 2, \\ 3\varepsilon/2 & \text{if } j = 0 \text{ or } j = |B_\varepsilon| - 1. \end{cases} \tag{52}$$

This implies that, by independence of the $P(0|z_1^k)$'s across $z_1^k \in \mathcal{X}^k$,

$$H_{\text{Shannon}}\left(\left\{P(0|z_1^k) : z_1^k \in \mathcal{X}^k\right\}\right) \ge |\mathcal{X}|^k(1 - 3\varepsilon)\log(1/\varepsilon). \tag{53}$$

Let $e(z_1^k)$ be the random variable $P(0|z_1^k) - f(z_1^k)$. $P(0|z_1^k)$ is a measurable function of $f(z_1^k)$ and $e(z_1^k)$, and therefore,

$$H_{\text{Shannon}}\left(\left\{f(z_1^k) : z_1^k \in \mathcal{X}^k\right\} \cup \left\{e(z_1^k) : z_1^k \in \mathcal{X}^k\right\}\right) \ge H_{\text{Shannon}}\left(\left\{P(0|z_1^k) : z_1^k \in \mathcal{X}^k\right\}\right) \tag{54}$$

Note that $e(z_1^k)$ is bounded in the rate $\{-r, \cdots, r\}$ and can take at most $2r + 1$ values. Therefore, $H\left(\left\{e(z_1^k) : z_1^k \in \mathcal{X}^k\right\}\right) \le |\mathcal{X}|^k \log(2r + 1)$. Since $H(A, B) \le H(A) + H(B)$, we have that,

$$H_{\text{Shannon}}\left(\left\{f(z_1^k) : z_1^k \in \mathcal{X}^k\right\}\right) \ge H_{\text{Shannon}}\left(\left\{P(0|z_1^k) : z_1^k \in \mathcal{X}^k\right\}\right) - H\left(\left\{e(z_1^k) : z_1^k \in \mathcal{X}^k\right\}\right) \tag{55}$$

$$\ge |\mathcal{X}|^k\left((1 - 3\varepsilon)\log(1/\varepsilon) - \log(2r + 1)\right). \tag{56}$$

$\square$

Recall that we are guaranteed that $g(x_{t+k+1} = 0, H_{t+k}) \in P(0|z_1^k) + [-\varepsilon - \eta_P(t), \varepsilon + \eta_P(t)]$. This implies that the recurrent model is able to recover $\text{round}(p)$ for $p = P(0|z_1^k)$ up to an error of $r = \lceil \eta(t)/\varepsilon \rceil$ for each $z_1^k \in \mathcal{X}^k$ by computing $\text{round}(\hat{p})$ where $\hat{p} = g(x_{t+k+1} = 0, H_{t+k})$. Informally, this just means that $\hat{p}$ is likely to fall in a bucket close to $p$. In combination with Lemma 1, for $f(z_1^k) = g(x_{t+k+1} = 0, H_{t+k})$ we have that,

$$H_{\text{Shannon}}(\{g(x_{t+k+1} = 0, H_{t+k}) : z_1^k \in \{0,1\}^k\}) \ge 2^k\left((1 - 3\varepsilon)\log(1/\varepsilon) - \log(2\lceil \eta_P(t)/\varepsilon \rceil + 1)\right) \tag{57}$$

Note however, that $g(x_{t+k+1} = 0, H_{t+k})$ is a function of $z_1^k$ implicitly, through $H_{t+k}$ (which is also a function of $x_1^t$). Since the dimensionality of $H_{t+k}$ is $d$ and the model is implemented to $\texttt{p}$ bits of precision,

$$H_{\text{Shannon}}(\{g(x_{t+k+1} = 0, H_{t+k}) : z_1^k \in \{0,1\}^k\}) \le H_{\text{Shannon}}(H_{t+k}) \le d\texttt{p} \tag{58}$$

where all randomness here is induced by the random draw of the $k^{\text{th}}$-order Markov kernel $P$. Therefore, for the correctness guarantee Equation (45) to hold, we need,

$$d\texttt{p} \ge 2^k\left((1 - 3\varepsilon)\log(1/\varepsilon) - \log(2\lceil \eta_P(t)/\varepsilon \rceil + 1)\right) \tag{59}$$

in the limit $t \to \infty$, and noting that $\limsup_{t\to\infty} \eta_P(t) = 0$ almost surely over $P$ drawn from the prior, it is necessary that,

$$d\texttt{p} \ge 2^k(1 - 3\varepsilon)\log(1/\varepsilon). \tag{60}$$

$\square$

Fig. 6 provides empirical evidence that the exponential dependency of the theorem on the order $k$ is consistent.

**Remark.** The proof above assumes that the $k^{\text{th}}$-order Markov chain is on a binary state space. However, the result can easily be extended to give the lower bound $d \cdot \texttt{p} \ge \Omega(|\mathcal{X}|^k)$ for larger state spaces, as well as similar scaling results for priors $\text{Dir}(\beta \cdot \mathbf{1})$ for any $\beta > 0$. Furthermore, we believe it should be possible to replace the $L_\infty$ error guarantee in Equation (6) by the KL-divergence between the two distributions without significantly changing the conclusion ($d \cdot \texttt{p} = 2^{\Omega(k)}$).

**Intuition.** The intuition behind this result is in the manner in which the recurrent architecture carries out computation: by proceeding sequentially and compressing the information from the sequence it has seen thus far at some time $t$ into a small hidden vector, the model does not know what the next

$k$ tokens will be: the knowledge of this is vital to be able to compute the add-$\beta$ estimator at time $t + k + 1$ with a small memory footprint. Indeed, when the identity of the next $k$ tokens changes, the output of the model at time $t + k + 1$ must look drastically different (as the add-$\beta$ estimator corresponds to approximately evaluating $\mathbb{P}(\cdot|i_1^k)$, which are unrelated distributions under different choices of $i_1^k$). There are $\sim 2^{2^k}$ possible values the set $P = \{\mathbb{P}(\cdot|i_1^k) : i_1^k \in \{0,1\}^k\}$ can take. But when $d$ and p are small, the output of the model just cannot take so many values: it can realize at most $2^{d\mathrm{p}}$ possible sets. In other words, in order to succeed, the recurrent architecture is essentially forced to keep track of the number of occurrences of each $i_1^k \in \{0,1\}^k$ in the sequence at each time $t$, which costs an exponential dependence on $k$ in the hidden dimension/precision.

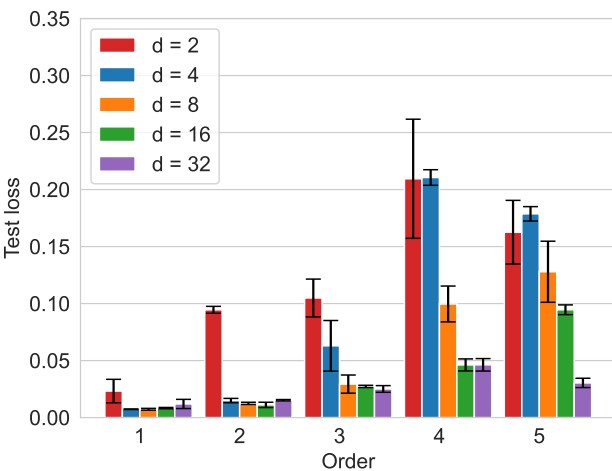

Figure 6: Relation between the Markov order $k$ and the hidden dimension $d$ of the 1-layer Mamba model. The plot shows that $d = 2^k$ is sufficient for the model to learn the $k$-th order Markov task. This corroborates the fact that Theorem 2 in the main paper has the correct order dependency.

# E    ADDITIONAL RESULTS

## E.1    LINEAR ATTENTION

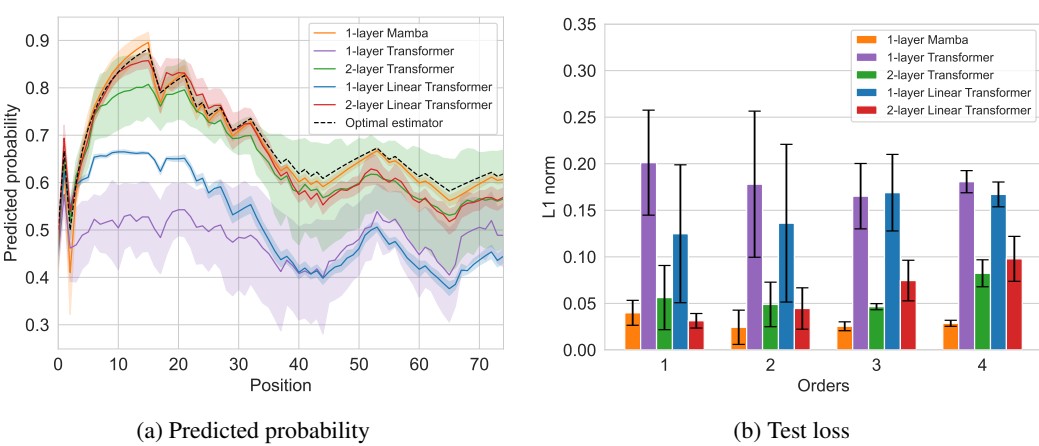

(a) Predicted probability                                    (b) Test loss

Figure 7: Comparison of predicted probability and test loss between a 1-layer Mamba and the other baselines, including linear attention. Mamba outperforms all baselines. Linear attention and softmax attention Transformers perform similarly.

## E.2    MULTIPLE STATES

Fig. 8 shows that our results extend to larger number of states.

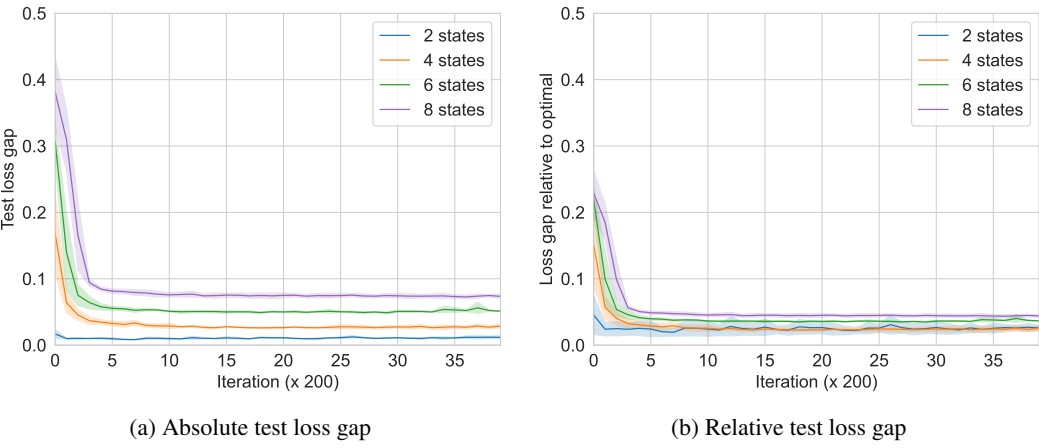

(a) Absolute test loss gap                                    (b) Relative test loss gap

Figure 8: Test loss gap from the optimal for 1-layer Mamba and first-order Markov data, for different number of states. (a) shows the absolute gap $L(\theta) - L^*$; (b) shows the relative gap $(L(\theta) - L^*)/L^*$. The loss gap is consistently small for all state sizes.

## E.3    DEEPER NETWORKS

Fig. 10 show that our results are consistent with larger number of layers.

## E.4    OVER-PARAMETRIZED SETTINGS

Fig. 9 shows that our observations hold also for larger convolution width and deeper networks.

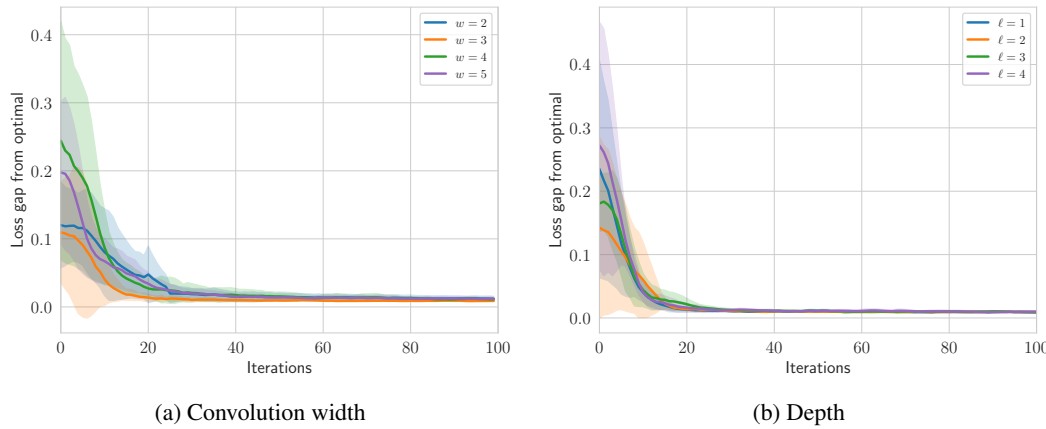

(a) Convolution width          (b) Depth

Figure 9: Test loss gap from optimal across iterations in over-parametrized settings. (a) shows 1-layer Mamba with varying convolution width. (b) shows a Mamba with convolution width $w = 2$ and varying number of layers. The models correctly learn the optimal predictor in all cases.

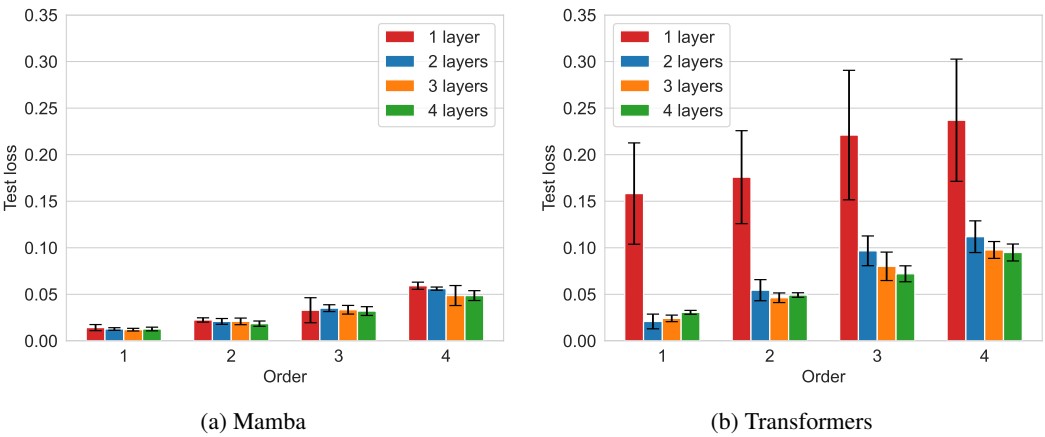

(a) Mamba          (b) Transformers

Figure 10: Test loss gap from the optimal for Mamba and Transformers, for different number of layers and Markov orders. Mamba has a smaller loss gap than transformers across all orders. Furthermore, adding more layers to Mamba does not significantly improve performance. As expected, 1-layer Transformers cannot solve the Markov task, while 1-layer Mamba can.

### E.5 HOLD-OUT EXPERIMENT

We also consider the following interesting experiment: we trained Mamba only on sequences from a subset of the Markov simplex, specifically the interval $[0, 0.5]$, and we tested it on sequences from its complement $[0.5, 1]$. Interestingly, our experiments show that the test loss still converges to the optimal. However, by inspecting the actual estimated probabilities on a fixed test sequence, we see that the absolute difference between the model's estimation and the optimal Laplacian smoothing is high for the first samples, and gradually decreases as the sequence progresses. This is justified by the fact that the Laplacian estimator is not only Bayes-optimal, but also minimax optimal (i.e., independently of the prior on the simplex) in the limit of long sequences (i.e., the gap from the optimal loss goes to 0 as $n \rightarrow \infty$). Table 2 shows the absolute difference $|\mathbb{P}_\theta(x_t = 1|x_1^{t-1}) - \mathbb{P}^*(x_t = 1|x_1^{t-1})|$ for several $t$, showing how this difference gradually goes to zero as $t$ increases.

### E.6 CONVOLUTION

Fig. 11 shows that adding convolution to transformers (similarly to Mamba) makes the model solve the task with just one layer. On the contrary, Fig. 12 shows that Mamba needs two layers to solve the

Table 2: Results for the hold-out experiment.

| Length $t$ | Abs. difference from optimal |
|:---:|:---:|
| 1 | $0.378 \pm 0.050$ |
| 50 | $0.098 \pm 0.039$ |
| 100 | $0.007 \pm 0.003$ |
| 300 | $0.001 \pm 0.001$ |

task if convolution is removed. Table 3 shows that a one-layer Mamba without convolution cannot learn the optimal estimator, no matter how wide the model is.

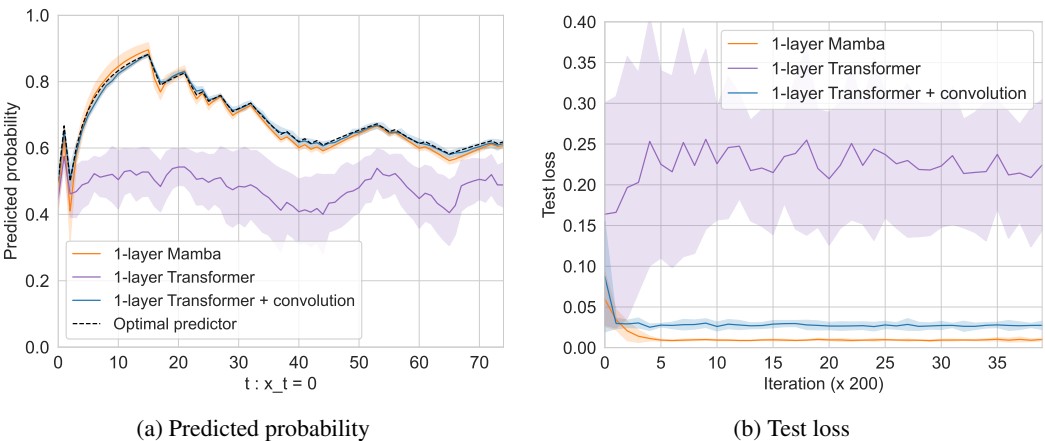

(a) Predicted probability         (b) Test loss

Figure 11: Predicted probability and test loss for Transformers with and without convolution. Adding convolution to the $K, Q, V$ matrices of transformers makes the models succeed in learning the Markov task, similarly to 1-layer Mamba.

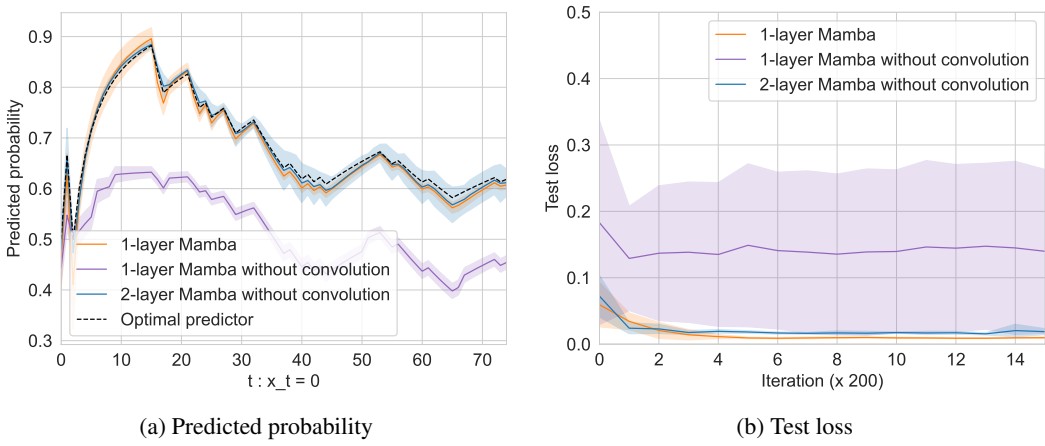

(a) Predicted probability         (b) Test loss

Figure 12: Predicted probability and test loss for the full 1-layer Mamba and a 2-layer Mamba without convolution. Similarly to transformers, Mamba needs two layers to solve the Markov task when convolution is removed.

## E.7 SWITCHING MARKOV

Here we detail the switching Markov process more formally:

1. Initialize $t = 0$.

Table 3: Experiments on one-layer Mamba without convolution, with varying width. The model does not learn the optimal estimator successfully.

| Hidden dimension $d$ | Avg. test loss gap from optimal |
|---|---|
| 10 | $0.113 \pm 0.032$ |
| 100 | $0.158 \pm 0.055$ |
| 1000 | $0.140 \pm 0.072$ |

2. Draw a binary Markov kernel $P$ with each row sampled i.i.d. from $\mathrm{Dir}(\beta \cdot \mathbf{1})$.
3. Let $x_t = S$ with probability $p_{\mathrm{switch}}$, or sample $x_t \sim P_{x_{t-k+1},:}^t$ with probability $1 - p_{\mathrm{switch}}$.
4. If $x_t = S$, set $t = t + 1$ and go to step 2; if $x_t \neq S$, set $t = t + 1$ and go to step 3.

Fig. 13 illustrates the behavior of Mamba on this process when $p_{\mathrm{switch}} = 0.01$.

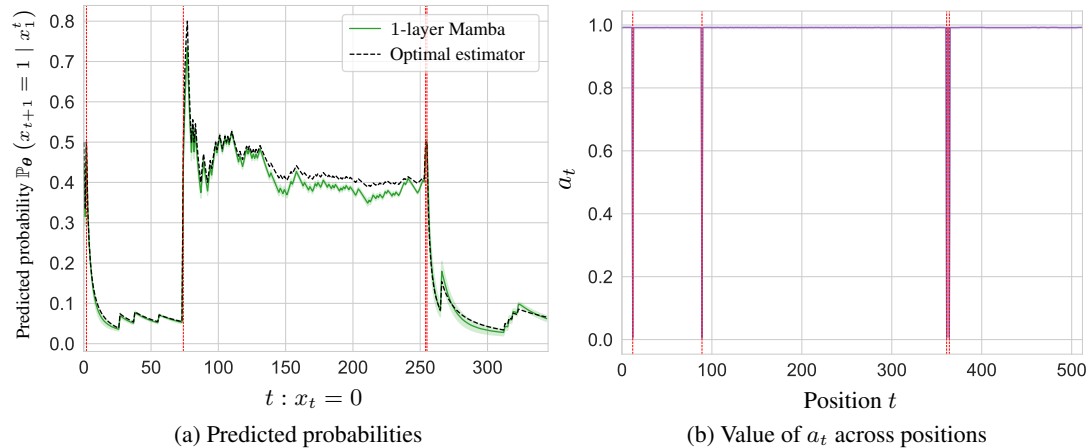

(a) Predicted probabilities

(b) Value of $a_t$ across positions

Figure 13: One-layer Mamba on switching Markov data. Mamba is able to learn the optimal predictor by forgetting past counts every time a switch token occurs. This is achieved by setting $a_t = 0$ at every switch, and $a_t = 1$ otherwise.

### E.8  FURTHER ABLATION FOR NATURAL LANGUAGE

The fundamental role played by convolution in modeling natural language is demonstrated in Table 1. However, we find that other components, in particular gating factor $a_t$, play an essential role as well, when natural language is considered. In Table 4, we show the increase in test perplexity when convolution, gating factor $a_t$ and non-linearities are individually removed from the full Mamba-2 architecture.

Table 4: Perplexity results on the WikiText-103 dataset.

| Model | Params. | Perplexity | Percentage increase |
|---|---|---|---|
| Mamba-2 (full) | 14.54 M | **27.55** | – |
| Mamba-2 (w/o conv) | 14.53 M | 30.68 | 11% |
| Mamba-2 (w/o gating factor) | 14.54 M | 32.16 | 17% |
| Mamba-2 (w/o non-linearities) | 14.54 M | 28.98 | 5% |

Table 5: Perplexity results on WikiText-103 and PG-19 datasets for Mamba with 12 layers.

| Dataset | Model | Params. | Perplexity |
|---|---|---|---|
| WikiText-103 | Mamba-2 | 110 M | 21.38 |
| WikiText-103 | Mamba-2 w/o convolution | 110 M | 21.46 |
| WikiText-103 | Mamba-2 w/o gating | 110 M | 21.71 |
| PG-19 | Mamba-2 | 200 M | 14.16 |
| PG-19 | Mamba-2 w/o convolution | 200 M | 14.28 |
| PG-19 | Mamba-2 w/o gating | 200 M | 14.66 |

# F  MODEL ARCHITECTURES AND HYPER-PARAMETERS

The following tables discuss details on the architectures and hyperparameters used in all the paper's experiments. Each experiment was run on a single Nvidia A100 GPU. The time taken by each experiment was between 10 to 60 minutes.

Table 6: Parameters in the Mamba architecture with their shape.

| Parameter | Matrix shape |
|---|---|
| embedding | $2 \times d$ |
| mamba.A | 1 |
| mamba.dt | 1 |
| mamba.in_proj | $(2ed + 2N + 1) \times d$ |
| mamba.conv1d | $(ed + 2N) \times w$ |
| mamba.out_proj | $d \times (2ed + 2N + 1)$ |
| mlp.fc1 | $4d \times d$ |
| mlp.fc2 | $d \times 4d$ |
| lm_head | $d \times 2$ |

Table 7: Settings and parameters for the Mamba model used in the experiments.

| | |
|---|---|
| Dataset | $k$-th order binary Markov source |
| Architecture | Based on the Mamba-2 architecture as implemented in Dao & Gu (2024) |
| Batch size | Grid-searched in $\{16, 32, 64, 128, 256\}$ |
| Accumulation steps | 1 |
| Optimizer | AdamW ($\beta_1 = 0.9, \beta_2 = 0.95$) |
| Learning rate | 0.001 |
| Scheduler | Cosine |
| # Iterations | 10000 |
| Weight decay | $1 \times 10^{-3}$ |
| Dropout | 0 |
| Sequence length | Grid-searched in $\{128, 256, 512\}$ |
| Embedding dimension | Grid-searched in $\{2, 4, 8, 16, 32\}$ |
| Mamba layers | 1 |
| Heads | 1 |
| Convolution window | Between 2 and 6 |
| Repetitions | 5 |

Table 8: Parameters in the transformer architecture with their shape.

| Parameter | Matrix shape |
|---|---|
| transformer.wte | $2 \times d$ |
| transformer.wpe | $N \times d$ |
| transformer.h.ln_1 ($\times \ell$) | $d \times 1$ |
| transformer.h.attn.c_attn ($\times \ell$) | $3d \times d$ |
| transformer.h.attn.c_proj ($\times \ell$) | $d \times d$ |
| transformer.h.ln_2 ($\times \ell$) | $d \times 1$ |
| transformer.h.mlp.c_fc ($\times \ell$) | $4d \times d$ |
| transformer.h.mlp.c_proj ($\times \ell$) | $d \times 4d$ |
| transformer.ln_f | $d \times 1$ |

Table 9: Settings and parameters for the transformer model used in the experiments.

| | |
|---|---|
| Dataset | $k$-th order binary Markov source |
| Architecture | Based on the GPT-2 architecture as implemented in Pagliardini |
| Batch size | Grid-searched in $\{16, 32, 64, 128, 256\}$ |
| Accumulation steps | 1 |
| Optimizer | AdamW ($\beta_1 = 0.9, \beta_2 = 0.95$) |
| Learning rate | 0.001 |
| Scheduler | Cosine |
| # Iterations | 10000 |
| Weight decay | $1 \times 10^{-3}$ |
| Dropout | 0 |
| Sequence length | Grid-searched in $\{128, 256, 512, 1024\}$ |
| Embedding dimension | Grid-searched in $\{4, 8, 16, 32\}$ |
| Transformer layers | Between 1 and 2 depending on the experiment |
| Attention heads | 1 |
| Repetitions | 5 |

