# OpenReview forum: "From Markov to Laplace: How Mamba In-Context Learns Markov Chains"
_ICLR.cc/2026/Conference — ICLR 2026 Oral_

### Official Review · Reviewer_PRst · 2025-10-25

**Soundness:** 4
**Presentation:** 4
**Contribution:** 3
**Rating:** 8
**Confidence:** 3

**Summary:**

In this paper, the authors intended to provide theoretical understanding on the learning capability of MAMBA on Markovian data. The authors are able to demonstrate through a constructive proof that Mamba is able to efficiently learn the
in-context Laplacian smoothing estimator even with just one-layer, thus ensure both Bayes and minimax optimality through the properties of the Laplacian. In addition, the authors established the lower bound on the hidden dimension required for the representing the optimal estimator.

**Strengths:**

The paper is very well-written, the basic concepts and main steps in the arguments are clearly presented and explained.

The results presented in the paper is valuable to the understanding of Mamba.

**Weaknesses:**

The optimality of the Laplacian estimator is based on the Dirichlet distribution assumption, it will be helpful to see the impact of the assumption.

It would be helpful that the quality of the lower bound presented in Theorem 2 can be addressed.

**Questions:**

My main question is related to the Dirichlet distribution assumption and its relation with Laplacian estimator, How much do these assumptions affect the generality of the results.

---

> ### Author Response · Authors · 2025-11-23
>
> We thank the reviewer for their appreciation of our work and for their insightful questions. We address their major points below.
>
> **Dirichlet assumption**
>
> For the Markovian setting that we considered, the Dirichlet assumption is not essential, provided that the sequence length is large enough. In fact, it can be proved (see, e.g., [1]) that the Laplacian estimator is asymptotically (as the sequence length becomes large) minimax optimal, i.e., for all prior distributions on the Markov kernel. To test this fact, we conducted a hold-out experiment, where the model is trained only on half of the simplex and tested on the other half. The results show that the difference in prediction between Mamba and the optimal estimator quickly goes to zero as the sequence length increases, confirming the theory. The results are included in Appendix E.5.
>
> **Lower bound**
>
> While Theorem 2 holds in broad generality for all recurrent architectures, of which Mamba is just an example, verifying it empirically remains challenging due to the asymptotic nature of the result. In fact, for small orders, constants and numerical precision strongly affect the bound, while the exponential dependency on the order is dominant for larger $k$. Nonetheless, in the main paper we included an experiment on Mamba to empirically verify the minimum hidden dimension necessary to learn the optimal $k$-th order estimator, for $k \leq 5$. The results are shown in Figure 6 of the revised PDF. While they corroborate the exponential relation proved in Theorem 2, more definitive results would require testing larger orders, which unfortunately becomes quickly computationally infeasible due to the very large hidden dimension required.
>
> We hope to have addressed all the reviewer’s questions and we remain available for further discussion on our paper.
>
> ---
> **References**
>
> [1] Hao et al., “On Learning Markov Chains”, 2018.

---

> > ### Comment · Reviewer_PRst · 2025-11-26
> >
> > Thank the authors for their rebuttal.  My concerns are addressed, and I will maintain my evaluation.

---

> > > ### Author Response · Authors · 2025-11-27
> > >
> > > Dear Reviewer PRst, we are happy to have addressed your concerns. We thank you again for your appreciation of our work.

---

### Official Review · Reviewer_6hFD · 2025-11-01

**Soundness:** 3
**Presentation:** 3
**Contribution:** 3
**Rating:** 8
**Confidence:** 3

**Summary:**

This paper builds upon the previous Markov-ICL framework to empirically and theoretically investigate the ICL ability of Mamba architecture.
Specifically, they design experiments to train Mamba for next-token prediction loss on random Markov chains, and find one layer Mamba match the performance of the statistical optimal estimator, and convolution layer play an important role to ahcieve this performance. Theoretically, the authors provide a constructive proof showing how Mamba's components work in concert to implement the laplace smoothingy: convolution identifies the $k^{th}$-order transition, recurrence (with $a_t \approx 1$) accumulates all transition counts into the hidden state, and selectivity reads out the specific counts needed for the current prediction. Finally, the paper proves a lower bound showing that for Mamba the hidden dimension $d$ must scale exponentially with the Markov order $k$.

**Strengths:**

1. This is the first work to investigate theoretically the ICL ability of the Mamba architecture.

2. The paper is well-written; the logic and arrangement are easy to follow, and the author provides enough background in the main text to help readers understand the scientific question being studied. It starts from an empirical observation, finds convolution plays an important role, and then echoes this importance in the theoretical analysis.

3. The paper is perfectly designed and written to shed light not only on the theoretical analysis but also on the real-world application and the Mamba architecture. The theory echoes the key component and mechanism in the original Mamba paper (convolution and the selective mechanism) and is supported by experiments on synthetic data.

**Weaknesses:**

1. One possible weakness is that, to the best of our current understanding, Mamba has not demonstrated the strong ICL ability seen in large transformer-based language models. While this is not a weakness of the theoretical analysis itself, Mamba's actual ICL capability in real-world applications will influence the impact of this theoretical work on the community.

2. The real-world experiment is only on the language modeling task (e.g., perplexity on WikiText-103) instead of a dedicated, real-world ICL experiment, which limits the generality of the findings.

**Questions:**

1. Does the result in Theorem 2 imply that the transformer is more powerful than Mamba in in-context learn the complex task (higher order Markov process)? Can we conclude transformer is more parameter-efficient than Mamba when solving complex tasks? Do authors have any insight into a way to design a Mamba block that sidesteps this trade-off?

2. In the theoretical proof, the author omits the gating mechanism in Mamba, but later found in the NLP task that gating is an important component. If we want to add gating back to theoretical analysis, how does it interact with the "counting" mechanism described in Theorem 1?

---

> ### Author Response · Authors · 2025-11-23
>
> We thank the reviewer for their appreciation of our work and for their insightful questions. We address their major points below.
>
> **Mamba’s ICL capabilities**
>
> We agree with the reviewer that Mamba’s ICL capabilities lack the strong empirical support obtained for Transformers. Our findings, however, show theoretically and empirically that Mamba does have the capacity to implement at least some types of ICL mechanisms, albeit quite differently from Transformers. The main issue might be that Mamba is much less parameter efficient in that, as shown by our Theorem 2. In fact, while Transformers can implement higher-order estimators with parameter dimensions growing linearly in the Markov order [1], Mamba requires exponentially many.
>
> **Real-world experiments**
>
> While we agree with the reviewer that real-world ICL experiments would be beneficial, our choice of the Markovian data model stems from the necessity to find a non-trivial setting whose optimal ICL solution is mathematically tractable, allowing for a rigorous analysis of the inner mechanisms of Mamba. We then tested our findings on real-world language modeling settings (WikiText-103) for further validation. We also run some additional language modelling experiments for larger 12-layer models; see our response to Reviewer ntkq and Table 5 in the revised PDF.
>
> **Comparison to Transformers**
>
> Indeed, our Theorem 2 proves that Mamba is much less parameter-efficient than Transformer in implementing higher-order estimators, as the former requires a number of parameters exponential in the order, while the latter only requires linearly many [1]. This may suggest that Mamba models are more efficient in learning short-term dependencies, while Transformers are more efficient for long-term ones. Unfortunately, this is an essential limitation of recurrent models that rely on a compressed hidden state. This cannot be overcome easily, e.g., by playing with non-linearities or increasing the number of layers. We believe that a solution is to design architectures that alternate attention and Mamba layers, and we already see many new architectures going in this direction.
>
> **Effect of gating**
>
> Regarding our theoretical construction in Theorem 1, gating can be easily included without affecting the overall computation. In fact, note that in our construction (Appendix C.2), $\mathbf{x}_t$ is zero everywhere except in the (even) coordinate $2x_t$ – $x_t\in \\{1,2,\dots,S\\}$, where it equals 1. Therefore, one can simply construct $W_z$ in Equation (Mamba) such that $W_z \mathbf{x}_t$ is the all-one vector, effectively leaving $\mathbf{y}_t = H_t \mathbf{c}_t$ unaffected.
> While not necessary for our specific case, in general we believe that gating can play a very useful role in facilitating similar mechanisms as the one we observed in our paper. In fact, gating can nullify some coordinates based on the current input. This can be very useful in maintaining only the information (e.g., in our case, the counts) that is relevant for the current token, and cancelling out the rest.
>
> We hope to have addressed all the reviewer’s questions and we remain available for further discussion on our paper.
>
> ---
> **References**
>
> [1] Rajaraman et al., “Transformers on Markov data: Constant depth suffices”, 2024.

---

### Official Review · Reviewer_5agm · 2025-11-02

**Soundness:** 3
**Presentation:** 3
**Contribution:** 4
**Rating:** 8
**Confidence:** 2

**Summary:**

This paper studies the in-context learning abilities of the Mamba (S6) architecture, as a cheaper alternative to transformers (specifically, their quadratic attention cost in the sequence length). To study this problem, the authors consider finite-state k-order Markov Chains, and evaluate Mamba and Transformers in their ability to output the transition probabilities of the optimal Laplacian smoothing estimator in-context. Empirically, the findings include Mamba fitting the Laplacian smoothing estimator with only 1 layer, and convolution being the main ingredient in this ability. Furthermore, the authors show that a simplified Mamba variant can theoretically encode all the bits needed for storing the Laplacian smoothing operator with specific choices of its parameters.

**Strengths:**

- The paper is clearly written and well motivated.
- The methodology considered in this study is sound and made explicit: Building on the empirical observation that the convolution is the key ingredient in the ICL abilities of Mamba, the authors construct a minimal architecture that allows a theoretical analysis all while keeping key similarities with the original architecture. This allows the derivation of sound theoretical results characterizing Mamba's ability to learn in-context, as function of the choices in parameters and the hidden dimension.
- Given equivalences between language modeling, and k-order Markov Chains (e.g. n-gram models which previously had been the standard in language modeling), I believe that the insights included in this study are an important step towards building more efficient architectures in general, and in Language modelling in particular.

**Weaknesses:**

- Theorem 1 is insightful in proving the in-context ability of Mamba. However, in my understanding it only tackles the model misspecification problem, in the sense that it proves that the optimal solution can be represented by the model (zero approximation error). I think it would be interesting to discuss the intersections between the theoretically chosen parameters that achieve the result in Theorem 1, and those actually obtained by optimizing Eq 3 on a given training set (with or without distribution set) starting from standard random initializations of these parameters. This is partially done, with the observation that $a_t \approx 1$ at convergence, but can also be extended to other parameters (for instance, the count-independent term in Eq 5 that need to sum up to $\beta \mathbf{1}$).
- The natural language modeling experiments only validate the key role of the convolution in Mamba. I think that even the theoretical results (especially Th 2) can also be validated through experiments. For instance, by generating language from an n-gram model, one can show that a given hidden dimension size of Mamba is strictly needed in order to be able to perform in-context learning tasks in the generated language corpus. Of course this comes with challenges related to the exponential dependence on the Markov Chain order, but having a discussion around this would be a nice addition to the paper.

**Questions:**

- Did the authors check that 2 or more layers Mamba achieves the same result as 1-layer Mamba in the considered experiments? Or in other words, what is the impact on performance if the model is over-parameterized with respect to the bits needed to encode the Laplacian smoothing estimator?

---

> ### Author Response · Authors · 2025-11-23
>
> We thank the reviewer for their appreciation of our work and for their insightful questions. We address their major points below.
>
> **Empirical validation of the parameters**
>
> Following the reviewer’s suggestion, we run some experiments to further validate empirically our theoretical construction. The results are shown in Figure 5 in the updated PDF. Specifically, we evaluated the inner products $\mathbf{b}^{(ij)\top} \mathbf{c}^{(k)}$ across training and verified that the incorrect counts corresponding to $i\neq k$ all have inner products going to zero at convergence. This validates our construction according to which only the correct counts involved in the computation of the optimal estimator are kept in the final logit. We also evaluated the count-independent vector across iterations for the binary case. As Figure 5(b) shows, both coordinates converge to $\beta=1$, which is optimal.
>
> **Empirical validation of Theorem 2**
>
> As the reviewer pointed out, verifying Theorem 2 empirically is challenging due to the asymptotic nature of the result. In fact, for small orders, constants and numerical precision strongly affect the bound, while the exponential dependency on the order is dominant for larger $k$. Nonetheless, in the main paper we included an experiment on Mamba to empirically verify the minimum hidden dimension necessary to learn the optimal $k$-th order estimator, for $k \leq 5$. The results are shown in Figure 6 in the main paper. While they corroborate the exponential relation proved in Theorem 2, more definitive results would require testing larger orders, which unfortunately becomes quickly computationally infeasible due to the very large hidden dimension required.
>
> **Over-parametrized setting**
>
> Following the reviewer’s suggestion, we tested Mamba in the over-parametrized setting in two ways: by keeping one layer and increasing the convolution width, and by keeping the convolution width fixed and increasing the number of layers. The results are shown in Figure 9 in the revised PDF. The results show a similar performance across all experiments, confirming that the model is able to correctly learn the optimal estimator in all the cases.
>
> We hope to have addressed all the reviewer’s questions and we remain available for further discussion on our paper.

---

### Official Review · Reviewer_ntkq · 2025-11-03

**Soundness:** 4
**Presentation:** 3
**Contribution:** 3
**Rating:** 6
**Confidence:** 4

**Summary:**

The paper studies in-context learning in Mamba-2 using Markov Chains, and shows an interesting finding of a single layer Mamba-2 model being able to very closely track the performance of the optimal in-context Laplacian smoothing estimator. The paper contains several experiments and ablations that focus on comparing a one layer Mamba-2 model to a 1 and 2 layer Transformer, specifically focusing on the Convolution part of the overall architecture as an important piece. The paper also presents several theoretical insights which map well to the observed results.

My review comes from viewing the paper from the lens of someone familar with the empirical results with larger scale Mamba-2 models and how they relate to Transformers at the whole model (not necessarily one or two layer wise) level.

Overall, I found the paper technically sound and addressing a relevant question to the Mamba line of models. I am slightly learning towards accepting the paper, but do have certain comments and questions that withhold me from a stronger rating (see below).

**Strengths:**

- I found the idea interesting! Thank you for sharing this from a perspective that is intriguing and different to the typical “we ran a bunch of experiments on the same benchmarks and boosted performance by X%” kind of LM study. It is positive to see Mamba models being connected with optimal statistical estimators.

- The paper is well written. Even though it may not be easy to read on a cursory look (I needed to go back and forth to see symbols and notation frequently), it does contain all the information necessary to understand it. The paper also introduces the core idea behind Mamba-2 well.

- In addition to the well motivated experiment operating on random Markov chains sampled from the vocabulary, the authors also repeat the same exercise on a true token distribution from a dataset like WikiText103 that would be more representative of the real world token transition probabilities. Modeling these “biased” next-token probabilities seems like the essence of language modeling and in-context learning to me, and I find it positive that we see perplexity results on real text too.

- Overall, I found the experiments to be well designed and aligned well with the theoretical insights.

**Weaknesses:**

- For the non-Markovian setting in Section 5, (ppl on WikiText103 being the only result), I feel like the evaluation could have been a bit more diverse and rigorous. This is the setting that I assume most practitioners (for better or worse) would be interested in, and it may have been useful to do this over a slightly wider range of model sizes (number of layers, dimensionality etc) to help build better intuition.

- There is a lot of related work on architectures that do sub-quadratic attention, e.g. the original Linear Attention work, or related recurrent/State Space style models like DeltaNet, RWKV etc. While I understand the specific focus on Mamba-2 in this work (which in itself needed a lot of analysis), the inclusion of 1-2 other related methods in the comparison (e.g. transformer with Linear Attn in addition to the 1 and 2 layer Transformer) would have strengthened my evaluation of the paper.

**Questions:**

- Re: the importance of convolution, I found these two related comments by the Mamba-2 author on their official implementation:
https://github.com/state-spaces/mamba/issues/525#issuecomment-2290739534
and
https://github.com/state-spaces/mamba/issues/433#issuecomment-2199557446
Specifically, “conv1d helps a bit on perplexity: e.g. at 360M on 7B tokens on the Pile, using GPT2 tokenizer, w conv1d we get around 8.6 perplexity and without conv1d is around 8.9 ppl.”. This represents a slight drop in overall model performance but not as significant as the one seen in the authors experiments in the paper. Do you have any theory about why this may be?

- In the Mamba-2 paper, the original authors illustrate the duality between Transformers and SSM’s. The Mamba-2 layer itself can be seen as a special case of a Transfomer (and vice versa) under certain assumptions and has generally very similar training characteristics. Given this duality, how do the authors feel about the presented results showing that a 1-layer Transformer is significantly worse at ICL than a one layer Mamba?

- Do you have any speculation about how your findings may change / evolve over multiple stacked layers as is the typical case in most models?

- For larger scale practical deployments of Mamba models, do you think we should expect better in-context learning results generally compared to Transformers?

---

> ### Author Response · Authors · 2025-11-23
>
> We thank the reviewer for their appreciation of our work and for their helpful suggestions to improve it. We address their major points below.
>
> **Importance of convolution**
>
> We would like to highlight an important fact: one Mamba layer can successfully approximate convolution (see, e.g., Proposition 3.2 in [1]). We think that this can help explain some of the important points raised by the reviewer, as follows.
> - _Difference in perplexity in experiments:_ We thank the reviewer for sharing these insightful comments on the role of convolution from Mamba’s authors, which we were unaware of. Notwithstanding the difference in the dataset and in the model parameters, we conjecture that the main reason for the difference in perplexity lies in the larger number of layers used by Mamba’s authors in their experiments compared to ours. In fact, since a Mamba layer can successfully approximate convolution, we conjecture that its importance diminishes when the number of layers becomes large, since its role can be taken over by some of the other layers of the network. Instead, when only a few Mamba layers are considered, convolution’s role becomes more prominent, as the other Mamba layers have to focus on implementing other tasks.
> - _Equivalence to Transformers:_ The equivalence between Mamba and Transformers, as proved in [2], refers exclusively to the pure Mamba layer, without accounting for convolution, which is essential for the implementation of the counting mechanism analyzed in our paper. Removing convolution from Mamba causes a drop in performance, similar to a one-layer Transformer (see Figure 12a of the revised PDF). Similarly, adding convolution to the $Q$, $K$ and $V$ matrices in a one-layer Transformer significantly improves its performance in implementing the optimal Laplacian estimator (Figure 11b), consistently with the equivalence shown in [2].
> - _Deeper networks:_ When a network with two or more Mamba layers is considered, there is no need for additional convolutions in order to implement the Laplacian estimator. In fact, the first Mamba layer can approximate convolution to differentiate the Markov contexts, which are then fed to the second Mamba layer for aggregation, with the same mechanism as described in our paper. Regarding the possibility of avoiding the exponential dependency on the Markov order unveiled by our Theorem 2, we point out that stacking more Mamba layers would not help. In fact, according to the general definition used in our Theorem 2, a $\ell$-layer recurrent model with $d$-dimensional hidden state can be interpreted as a (special case of) one-layer recurrent model with a $\ell \times d$-dimensional hidden state. This would lead to a lower bound that depends only linearly on $\ell$ and still exponentially on $k$.
>
> **More comprehensive experiments on language**
>
> We ran a few more experiments on real-world language datasets to provide a more comprehensive picture. Specifically, we ran two experiments with more layers/parameters (12 layers, 110-200M parameters) for two datasets, WikiText-103 and PG-19. The results are shown in Table 5 in the updated PDF. Both experiments show a drop in perplexity when removing either convolution or gating, albeit smaller in percentage compared to our original experiment. The results confirm that for natural language, both convolution and gating are important, showing a more nuanced picture than the Markov model. They also confirm our hypothesis that convolution is less important when deeper networks are considered. We included a discussion on this in Section 5.2 of the updated PDF.
>
> **Additional baselines**
>
> As suggested by the reviewer, we ran the experiment of the paper’s Figure 1 for an additional baseline: the original linear attention model. The results are shown in Figure 7 in the updated PDF, which we referenced in Section 3 in the main text. They show that linear attention performs very similarly to softmax attention, further confirming our findings.
>
> We hope to have addressed all the reviewer’s questions and we remain available for further discussion on our paper.
>
> ---
> **References**
>
> [1] Wang and Xue, “State-space Models with Layer-wise Nonlinearity are Universal Approximators with Exponential Decaying Memory”, 2023.
>
> [2] Dao and Gu, “Transformers are SSMs: Generalized Models and Efficient Algorithms Through Structured State Space Duality”, 2024.

---

### Author Response · Authors · 2025-12-03

Dear Reviewers and Area Chair,

As the discussion phase comes to an end, we would like to thank the reviewers once again for their unanimous appreciation of our work, and for their careful reading and insightful suggestions, which we believe have strengthened the paper even further. We are sorry that we could not engage further with them, due to the unexpected early termination of the rebuttal phase. Nonetheless, we hope to have addressed their questions thoroughly, and we believe to have implemented all of their suggestions in the revised manuscript. We would like to take this opportunity to summarize here all the changes to our paper that we made during the discussion phase:
- **Additional experiments on language:** following the suggestions of Reviewers ntkq and 6hFD, we included additional experiments on real-world datasets WikiText-103 and PG-19, which confirmed the important role of convolution even beyond the Markovian setting which is the main focus of our analysis. We also included a discussion on the less prominent role of convolution in very deep networks.
- **Additional baselines:** as requested by Reviewer ntkq, we included a comparison with an additional baseline, Linear Attention, which behaved similarly to softmax attention, aligning with our theoretical analysis.
- **Empirical validation of the theoretical construction:** following Reviewer 5agm, we included additional experimental verification of our theoretical construction, by probing several fundamental parameters across training. All the results further corroborated the agreement between our construction and Mamba’s learned mechanism in practice.
- **Over-parametrized setting:** as requested by Reviewer 5agm, we included additional experiments in over-parametrized settings, by increasing the number of layers and the convolution width of our model. The results confirmed the ability of Mamba to learn the optimal estimators in all conditions.

We thank the Reviewers and the Area Chair for their time and effort amid these unusual circumstances.

The Authors

---

### Meta-Review · Area_Chair_PEHh · 2026-01-15

**Summary:**

This paper investigates the in-context learning (ICL) capabilities of the Mamba-2 architecture through the lens of Markov Chains. The authors provide both empirical evidence and theoretical proofs demonstrating that a single-layer Mamba-2 model can approximate the optimal in-context Laplacian smoothing estimator. Key contributions include: 1) Identifying how Mamba's specific components (convolution, recurrence, selectivity) map to the steps of the Laplacian estimator (identifying context, accumulating counts, reading out probabilities). 2) A theoretical result showing that for recurrent models like Mamba, the hidden dimension must scale exponentially with the Markov order $k$, whereas Transformers scale linearly. 3) Highlighting the critical role of the convolution layer in shallow Mamba models for this task. Reviewers were overwhelmingly positive, praising the novelty of the theoretical analysis for Mamba, the clear connection between architectural components and statistical estimation, and the rigorous experimental design.

**Reviewer Concerns:**

Addressed by Rebuttal:

- Role of Convolution (Reviewer ntkq): The reviewer noted that in large-scale Mamba models, removing convolution has a negligible effect, contrasting with the paper's findings. The authors clarified that in deep models, Mamba layers can approximate convolution, whereas in the analyzed shallow setting, the explicit convolution component is critical. They also provided new experiments on deeper models (12 layers) on WikiText-103 to support this.

- Baselines (Reviewer ntkq): The authors added a Linear Attention baseline as requested, showing it performs similarly to softmax attention, reinforcing the paper's specific claims about Mamba.

- Optimization vs. Representation (Reviewer 5agm): The reviewer questioned if the model learns the optimal parameters or just can represent them. The authors provided empirical evidence showing the relevant inner products converge to zero, validating that the optimization process finds the theoretical construction.

- Dirichlet Assumption (Reviewer PRst): The authors clarified that the Laplacian estimator is asymptotically minimax optimal regardless of the prior, supported by a new hold-out experiment.

Outstanding:

- Real-world ICL Gap (Reviewer 6hFD): The reviewer noted that Mamba has not demonstrated the same strong ICL capabilities as Transformers in complex real-world settings. While the authors acknowledge this, this remains a limitation of the architecture that the paper characterizes but does not solve.

- Validation of Lower Bound for Large $k$ (Reviewers 5agm, PRst): Validating the exponential lower bound for high Markov orders is computationally infeasible. The authors validated it for small $k$, which was accepted by reviewers, but the high-$k$ regime remains theoretical.

**Reviewer Scores:**

Reviewer ntkq (Score: 6 -> 8): The reviewer was positive but withheld a higher score due to questions about the necessity of convolution in deep models and missing baselines. The rebuttal addressed both extensively (new baselines, experiments on deeper models).

Reviewer 5agm (Score: 8 -> 8): The reviewer was already a strong accept. The rebuttal addressed the optimization question.

Reviewer 6hFD (Score: 8 -> 8): The reviewer was a strong accept, valuing the theoretical first-principles approach.

Reviewer PRst (Score: 8 -> 8): The reviewer explicitly confirmed that their concerns regarding assumptions were addressed and maintained their high score.

---

### Decision · Program_Chairs · 2026-01-26

Accept (Oral)